# Plant-Derived Anti-Cancer Therapeutics and Biopharmaceuticals

**DOI:** 10.3390/bioengineering12010007

**Published:** 2024-12-25

**Authors:** Ghyda Murad Hashim, Mehdi Shahgolzari, Kathleen Hefferon, Afagh Yavari, Srividhya Venkataraman

**Affiliations:** 1Department of Cell & Systems Biology, University of Toronto, Toronto, ON M5S 3B2, Canada; 2Dental Research Center, Hamadan University of Medical Sciences, Hamadan 65175-4171, Iran; 3Department of Biology, Payame Noor University, Tehran P.O. Box 19395-3697, Iran

**Keywords:** cancer, phytochemicals, PVNPs, VLPs, MABs, phototherapy, plant viruses

## Abstract

In spite of significant advancements in diagnosis and treatment, cancer remains one of the major threats to human health due to its ability to cause disease with high morbidity and mortality. A multifactorial and multitargeted approach is required towards intervention of the multitude of signaling pathways associated with carcinogenesis inclusive of angiogenesis and metastasis. In this context, plants provide an immense source of phytotherapeutics that show great promise as anticancer drugs. There is increasing epidemiological data indicating that diets rich in vegetables and fruits could decrease the risks of certain cancers. Several studies have proved that natural plant polyphenols, such as flavonoids, lignans, phenolic acids, alkaloids, phenylpropanoids, isoprenoids, terpenes, and stilbenes, could be used in anticancer prophylaxis and therapeutics by recruitment of mechanisms inclusive of antioxidant and anti-inflammatory activities and modulation of several molecular events associated with carcinogenesis. The current review discusses the anticancer activities of principal phytochemicals with focus on signaling circuits towards targeted cancer prophylaxis and therapy. Also addressed are plant-derived anti-cancer vaccines, nanoparticles, monoclonal antibodies, and immunotherapies. This review article brings to light the importance of plants and plant-based platforms as invaluable, low-cost sources of anti-cancer molecules of particular applicability in resource-poor developing countries.

## 1. Introduction

Cancer is one of the most lethal diseases affecting humans due to its capability to metastasize, failure to treat and manage the disease appropriately, and lack of comprehensive knowledge of development mechanisms associated with cancer. Currently, the basic approaches in the treatment of cancer include chemotherapy, radiotherapy, and surgery. Such strategies do provide efficacy in the treatment of patients having early-stage cancers and non-metastatic cancers. However, they are by and large ineffective in achieving long-lasting beneficial effects in patients suffering from late-stage cancers and are often burdened with major impediments, such as the elicitation of resistance rendering chemotherapy ineffective in the long run, high toxicity associated with prolonged use of chemotherapy [1], and the occurrence of severe adverse side-effects caused by overall radiation doses as well as the destructive impacts of cytotoxic drugs on healthy human cells and physiological functions [2]. In such a context of increased prevalence of cancer and augmented burden from a socioeconomic viewpoint, the need for providing alternate cancer prevention and therapeutic approaches is compellingly urgent [https://www.who.int/health-topics/cancer#tab=tab_1; accessed on 20 October 2024].

Despite environmental exposure of the human population to various chemical carcinogens, a diet rich in vegetables and fruits forms a favored source of phytochemicals that preclude cancer. Phytochemicals confer properties such as detoxification, free radical scavenging, and antioxidant activities, which provide potent anti-cancer effects. Phytochemicals regulate cancer proliferative and cancer cell apoptotic pathways besides functioning as modulators of epigenetic mechanisms, leading to chemoprevention of cancer [3,4,5]. Medicinal plants are promising as adjuvants to augment the efficiency of anti-cancer drugs mediated through targeting of multiple signaling pathways while minimizing detrimental side effects [6,7]. Therefore, there is a compelling need to investigate cellular pathways and explore their cross-link with phytochemicals in cancer therapy. Phytochemicals control multiple signaling networks by upregulation of tumor suppressor genes, inhibition of oncogenes and modulating the expression of upstream and downstream mediators to preclude cancer progression.

The burgeoning immunotherapy technology has shown tremendous potential in averting or suppressing the progression of cancer, owing to its direct impact on malignant cancer cells with high efficacy to target and destroy cancer cells [8]. Recent investigations on antibody-enabled killing effects on tumor cells and the production of numerous antibodies against cancer cells has resulted in the generation of monoclonal antibodies (MABs) that particularly recognize specific antigens on the cancer cell surface. However, despite their benefits and treatment potential, their manufacture is not cost-efficient considering the requirements of high purity and quality of these antibodies in addition to the circumscribed scalability of the mammalian expression systems which impair their widespread use as anticancer therapeutics. Further, MABs produced in mammalian cells contain several mouse proteins and contaminants such as human pathogens. Plants provide a novel approach for generation of monoclonal antibodies against cancer. Plants inherently have high expression capability, inexpensive cultivation processes on large scales, lowered downstream processing steps subject to containment conditions while being bereft of human pathogens and ethical concerns associated with systems such as transgenic animals [9,10]. Hence, plants have tremendous potential as practically unlimited sources of MABs, referred to as “plantibodies”.

Oncolytic viruses are of high clinical value for their efficacy against cancer. However, the medical use of these viruses is challenged by the likelihood of reversion mutations of attenuated oncolytic viruses into their virulent forms as well as the possibility of integration of genomic sequences of the virus into the host genome [11]. On the other hand, plant viruses are not capable of infecting mammalian cells and therefore lack the above drawbacks related to infection and afford a valuable tool for manipulation of tumors and induction of anti-tumor immunity [11,12]. Plant virus nanoparticles do not replicate in or destroy cancer cells in a direct manner but constitute a novel class of immunostimulatory agents [12]. There are two types of plant viruses, namely whole viruses referred to as viral nanoparticles (VNPs) and virus-like particles (VLPs). VLPs are genome-free equivalents of VNPs, lack the ability to replicate in plants, and resemble the native structures of plant viruses. Both VLPs and VNPs can function as immune adjuvants and delivery systems for tumor-specific antigens which can be recognized by the human immune system. VNPs and VLPs are efficaciously taken up by antigen-presenting cells and can elicit strong immune responses. They have been used in cancer immunotherapy by direct injection into the tumors in order to induce anti-tumor immunity through the disruption of local immunosuppression, which renders support to locals, followed by systemic immunity against the tumor, a process referred to as “in situ vaccination”.

A recent review on plant-based platforms inclusive of plant-derived monoclonal antibodies, VNPs, and phytochemicals to treat cancer is lacking. This comprehensive review aims to provide insights into the molecular mechanisms underlying the anticancer activities of plant-derived bioactive compounds/phytochemicals and their integration into phototherapeutic strategies. Further, it discusses the use of plant-based monoclonal antibodies and plant viral nanoparticles in combating cancer.

## 2. Phytochemicals in Cancer Prevention and Therapeutics: Recent Developments

Cancer continues to impose a formidable global health challenge, demanding a perpetual quest for innovative and effective therapeutic approaches. Anticancer plants, a captivating realm within the broader spectrum of medicinal flora, represent a rich source of bioactive compounds that have demonstrated remarkable potential in the fight against cancer. These plants, often celebrated in traditional medicine practices, harbor an array of phytochemicals, such as alkaloids, flavonoids, polyphenols, and terpenoids, which manifest diverse biological activities with pronounced anticancer properties. Exploring these natural compounds has opened avenues for innovative and targeted therapeutic approaches in cancer treatment.

## 3. Major Benefits of Phytochemicals in Comparison with Synthetic Anti-Cancer Drugs

Chemotherapy is a therapeutic regime involving administration of combinations of synthetic drugs to the body. Significantly, this is the only one among a few therapeutic choices available to treat advanced stage or metastasized cancer. Nevertheless, an adverse drawback of chemotherapy is its lack of target selectivity [13]. Since cancer cells emerge from functional normal cells exhibiting unbridled growth, anticancer drugs show indiscriminate targeting of the growth and development of non-proliferative normal cells while inhibiting the growth of cancer cells. Such poor selectivity of most chemotherapeutic drugs leads to serious side effects impacting normal tissues, including hair follicles, gastrointestinal tract, and bone marrow [13]. Further, chemotherapeutic drugs compromise fertility and induce long-term damage to organs and even cause cancer. Additionally, these drugs pose toxicity issues to normal cells, leading to several side effects, of which some could be life-threatening. Many synthetic anticancer drugs have been shown to be associated with notable undesirable and adverse side effects, such as ototoxicity due to the long-term administration of cisplatin [14], cardiotoxicity due to doxorubicin [15], and cognitive impairment due to 5-fluorouracil [16]. These side effects include fatigue; pain; problems with the oral mucosa, skin, nail, and hair; anemia; nausea; dietary issues; and weight change. Also, there occur other side effects which may not be as potent but severely circumscribe the patients’ quality of life, leading to premature cessation of chemotherapy.

Synthetic anticancer drugs have poor solubility, diminished absorptions, and decreased oral availability [17]. Also, other complications, such as elicitation of multidrug resistance (MDR), emerge as these drugs are targeted to the DNA of a given cell wherein mutations occur, and, hence, the cell develops resistance. It is because of the emergence of MDR that cancer cells continue to grow despite the administration of chemotherapeutic drugs. Manifestations of drug tolerance in cancer cells include changes in the potential drug target or enhancement of mechanisms for cell survival, including alterations in apoptotic cycles caused by ceramide level changes, occurrence of DNA repair, inefficient p53 tumor suppressor protein or impact on cytochrome oxidases critical for cellular respiration. MDR also causes hyper-expression of efflux transporters based on the ATP binding cassette that in turn decrease the drug levels within the intracellular space to levels that are suboptimal [18]. Therefore, the adverse side effects of chemotherapy in addition to MDR development and the circumscribed therapeutic index of chemotherapeutic drugs severely impact the therapeutic efficiency of chemotherapy. Such side effect severity requires a reduction in the dosage of anticancer drugs, eventually leading to ineffectual treatment outcomes and potentiating metastasis. The effectiveness of synthetic anticancer drugs is circumscribed by complications such as frequent tumor relapse and initiation of metastasis [19]. Also, conventional chemotherapeutics drugs are prohibitively expensive. The delay in cancer diagnosis combined with non-responsive therapy causes high rates of mortality among several cancer patients. Approximately, 40.5% of men and women have been diagnosed with cancer at some point during their lifetimes (based on 2017–2019 data). In 2024, an estimated 14,910 children and adolescents aged 0 to 19 will be diagnosed with cancer, and 1590 will die of the disease [https://www.cancer.gov/about-cancer/understanding/statistics#:~:text=Approximately%2040.5%25%20of%20men%20and,will%20die%20of%20the%20disease; accessed on 20 October 2024].

Such a situation has instigated the compelling necessity to look for alternative anticancer drugs sourced from plant systems.

Phytochemicals have been found to be less toxic, specific, and selective to cancer cells due to which they show great promise as anticancer drugs [20,21,22,23]. Phytochemicals preclude DNA damage, promote DNA repair, slow down cancer cell growth, regulate hormones, and prevent the reproduction of damaged cells. They possess antioxidant properties while neutralizing free radicals that damage the DNA. Phytochemicals have been shown to suppress pathways involved in promoting cancer and cancer progression, eventually inhibiting unbridled cell growth and causing cell death through apoptosis. They alter proteins associated with various signal transduction pathways and exert definitive chemotherapeutic and chemopreventive roles by integrating with specific molecular signals. They execute multiple biological functions, such as antimutagenic, antiproliferative, antimetastatic, anti-angiogenesis, anti-inflammatory, antioxidant, and immunomodulatory properties, due to which they control cancer progression and intervene in different stages of cancer cell development. Moreover, they are involved in the regulation of the cell cycle as well as microRNA and lead to cancer cell death by promoting apoptosis and autophagy through ROS signaling. Chemopreventive phytochemicals substances vary in their efficacies based on the genotypes of individuals in the population. Hence, the combined administration of many phytochemicals contained in plant foods would enable suppression of carcinogenesis, providing synergistic or additive effects against cancer when compared to treatment with single phytochemicals [24].

## 4. Molecular Insights into the Roles of Phytochemicals Against Cancer

Phytochemicals used to preclude cancer are categorized as tumor blockers or tumor suppressors wherein blockers target events involved in cancer initiation, while suppressors preclude tumor promotion [25,26,27]. Several natural compounds, such as polyphenols including curcumin, flavonoids, lignans, stilbenes, tannins, and coumarins, as well as glucosinolates and isothiocyanates, block tumorigenesis by inducing the biosynthesis of detoxification enzymes [26,28], preventing DNA damage, genetic instability, and mutation, which instigate neoplastic transformation of cells. Such phase II detoxifying enzyme induction has been shown to be generated by Nrf2-ARE pathway activation [25,26,28].

On the other hand, suppressor agents inhibit tumor growth, development, and tumor progression in cells that have already been transformed and promote removal of cancer cells within the tumor mass. Such suppressing phytochemical agents include isothiocyanates found in cabbage, cauliflower, broccoli, and watercress; flavonoids occurring in citrus fruit, grapefruits, blueberries, and parsley; as well as coumarins sourced from cinnamon, tonka beans, and sweet clover [25,28]. These suppressors hinder tumor promotion via several mechanisms, including cell cycle arrest, angiogenesis inhibition, and repression of pathways supporting cancer cells, including NFkB [25,28].

Phytochemicals induce apoptosis by repressing the release of Bcl-2, Bcl-xL, and other anti-apoptotic proteins, stimulating the release of Bak, Bax, and other pro-apoptotic proteins that lead to cytochrome C release from the inner membrane of the mitochondria and apoptosome formation, which results in activation and release of caspases 3, 6, and 7 effector proteins. This degrades intracellular proteins and leads to nuclear fragmentation, blebbing, and cell shrinkage. Likewise, suppressors block the synthesis of hormones and hormone receptors within breast cancer cells that rely on hormones for tumor growth and development [25,26,28,29]. Phytochemicals are efficacious, widely available, non-toxic, and possess many biological activities, such as anticancer, pro-apoptotic, antioxidant, anti-proliferative, anti-angiogenic, and anti-inflammatory properties [30,31,32]. Table 1 enlists the most important phytochemicals and their anti-cancer activities against various types of cancers.

## 5. Potential for Cancer Therapy Based on Plant Characteristics: A Few Examples

*Dicoma anomala*, a perennial herb often called fever or stomach bush, belongs to the Asteraceae family. It contains an erect stem laden with thin hairs and has a tuber underground [135]. It is traditionally used in Africa for its medicinal properties. It contains phytochemicals such as triterpenes, sesquiterpenes, phytosterols, phenolic acids, flavonoids and acetylenic compounds occurring mainly in roots and leaves. It is used in the treatment of several cancers including breast, ovarian, kidney and prostate cancers [136,137]. Roots of *D. anomala* Sond show antiproliferative activities towards MCF-7 breast cancer cells in which sesquiterpene in conjugation with silver nanoparticles demonstrate anticancer properties by eliciting oxidative damage within the cancerous cells [138]. Aqueous extracts of *Dicoma capensis* have been shown to possess anticancer effects against MCF-7, MDA-MB-231, and MCF-12A breast cancer cell lines [139].

*Tribulus terrestris* is a perennial herb known for its anticancer activities attributed to its elevated levels of steroidal saponins that has been demonstrated to elicit programmed cell death in the cancerous MVF-7 cells via both intrinsic and extrinsic pathways of apoptosis [140,141,142]. Fruit extracts of *T. terrestris* inhibit autophagy in SAS and TW2.6 oral cancer cells and impact the growth, proliferation, migration as well as invasion of metastatic/neoplastic cancer cells [143].

*Withania somnifera* occurs as an evergreen, bushy shrub belonging to the Solanaceae family. Root extracts of this plant exhibit properties such as inhibition of vimentin found in areas of metastasis substantiating its anticancer effects in breast cancer [144]. The anticancer properties of *W. somnifera* have been primarily attributed to Withaferin A and Withanolide, two important phytochemicals found in this plant. These compounds modulate various signaling pathways, including reactive oxygen species (ROS), autophagy, and apoptosis [145]. *W. somnifera* extracts have been shown to restrict lung adenoma in male Swiss albino mice, and its root extracts demonstrate inhibition of ROS-elicited injury in mouse models [146]. Root ethanolic extracts inhibit A549 lung cancer cell proliferation by downregulating P13K which decreased metastasis [147]. In vitro effects of Withaferin-A include reduction in MDA-MB-231 human breast cancer cell proliferation mediated through the inhibition of TASK-3 potassium (K2P9) channel [148]. Further, this compound has been proved to block the proliferation of OKV-18 and SKOV3 ovarian cancer cell lines and HeLa, SKGII, and ME180 cervical cancer cell lines [149] by upregulating the p53 tumor suppressor along with arrest of cell growth as well as DNA damage signaling. Withanolides block MDA-MB-231 cell proliferation via the induction of apoptosis, Hsp70 overexpression, and reduction in ER expression in these cells [150]. Figure 1 shows the chemical structures of some important phytochemicals. Table 2 enlists the extraction methods popularly used to isolate and purify some of the major phytochemicals.

## 6. Extraction and Purification of Phytochemicals

For years, phytochemicals have functioned as an integral part of the global health system. Additionally, they have been used as lead compounds for synthesizing drugs. The extraction and purification of compounds from plants remains as the keystone of research concerning natural products. Hence, there has been a continual endeavor to identify improved extraction methods. Conventional extraction techniques include maceration, digestion, infusion and percolation, decoction, and Soxhlet extraction. However, conventional methods are associated with several disadvantages, such as protracted extraction times, use of significant quantities of solvents, lesser yields, compromising bioactivity, and often requiring a number of extraction steps. In addition, a large amount of phytochemicals that are thermolabile undergo degradation or decomposition during the heating process.

Therefore, the choice of extraction techniques determines the quality and reliability of subsequent analysis of the respective phytochemicals. The principal focus of the extraction methods is to attain shorter extraction times, better yields, environmental friendliness, and economic viability while not compromising their individual biological activities. In this context, many propitious modern and advanced green extraction techniques such as ultrasound, microwave, supercritical fluid, accelerated solvent, pressurized hot water extraction and enzyme-assisted extraction are gaining importance. Following extraction, their fractionation and purification are performed using many chromatographic methods such as thin-layer chromatography, paper chromatography, HPLC, and gas chromatography.

Reportedly, modern extraction methods have demonstrated several advantages compared to conventional techniques. Modern methods afford several favorable characteristics, such as decreased solvent demand, lesser time, better conservation of biological activities, improved yields, and diminished energy demand. Therefore, the choice of extraction and purification techniques is dependent on the plant matrix, the properties of the targeted phytochemicals, impacts on the environment, and economic viability.

Table 3 enlists some of the phytochemicals currently under clinical trials.

Taken together, several phytochemicals use various mechanisms to enable suppression of the survival and growth of cancer cells and target pathways such as PI3K/AktmTOR, JAK/STAT pathways, Hedgehog, Notch, and Wnt/β-catenin Hippo signaling pathways, leading to shut down of cancer cells followed by suppression of the heterogeneity, aggression, and remission of tumor cells [27].

## 7. Phytochemical-Based Nanoparticles in Cancer Prophylaxis and Therapy

The confluence of traditional healing wisdom and cutting-edge technology has given rise to phototherapy as a promising solution, offering targeted and minimally invasive strategies for the intricate battle against cancer [210]. Simultaneously, the rich reservoir of bioactive compounds inherent in various plant species has seized considerable attention due to its potential to mitigate the complex dynamics of cancer progression. Photo-mediated therapies such as photothermal therapy (PTT) and photodynamic therapy (PDT) have been found to be effectual in cancer therapy and operate by distinct damage mechanisms involving the production of heat and ROS, respectively [211,212], which results in cellular death. Therefore, both PDT and PTT can be applied to treat many types of cancer [213,214].

PDT involves the use of photosensitizer drugs that can be stimulated by radiation. PDT is scarcely or non-invasive and a very successful anticancer treatment strategy for several types of cancer [215]. PDT requires a light source, a photosensitizer (PS) drug and availability of oxygen whose interaction generates ROS [216]. Upon exposure to a specific wavelength of light, the PS absorbs photons, which results in conversion of the PS to its excited state as indicated in Figure 2. Subsequently, it crosses into a metastable triplet state that leads to type I PDT wherein the PS in its activated state can elicit a series of reactions involving biomolecules that produce radicals capable of reacting with molecular oxygen thereby generating ROS. In type II PDT, the PS directly transfers energy to oxygen, which leads to elicitation of ROS [217,218]. These ROS molecules have powerful oxidizing and cytotoxic effects.

Curcumin is a polyphenol known for its antitumor effects and photosensitizing characteristics [219]. Curcumin has been encapsulated within solid lipid nanoparticles towards use in phototherapy. These nanoparticles demonstrate augmented uptake of the drug into cancerous lung cells resulting in elicitation of ROS under exposure to light and this proved to be propitious towards phototherapy [220]. Nano-emulsions harboring curcumin as the photosensitizer drug were demonstrated to be greatly phototoxic towards breast cancer cells in addition to eliciting ROS at high levels [221]. Preparations of nano-emulsions containing acai oil were combined along with irradiation of light, which caused death of cancerous melanoma cells up to 85% [222]. This was further substantiated when the said nano-emulsion was shown to result in the reduction in tumor volume in murine models. A conjugate containing cyclodextrin and chlorophyll α when applied to colorectal human adenocarcinoma cells proved to be toxic to these cells under photo-induction, thus enabling PDT applications [223].

Another therapeutic strategy, PTT uses near-infrared laser/light (NIR) to elevate the temperature within the tumor site and elicits death of cancer cells [224]. Other sources of radiation to cause hyperthermia involve microwaves, visible light, ultrasound and radiofrequency waves [225]. PTT exhibits high specificity while being barely invasive [226]. The PTT strategy operates through two mechanisms: one includes the exposure of the tumor site to elevated temperatures (over 45 °C) for a few minutes, resulting in tumor cell death via thermal ablation, stasis within tumor vessels, as well as hemorrhage that preclude its administration with other treatment strategies; the other includes the induction of mild hyperthermia, wherein temperatures of 42–43 °C are set up that lead to tumor cell damage and augmented tumor vessel permeability that can be applied to promote the uptake of nanoparticles by tumors [212,227,228]. Tumor tissues show greater acidity and hypoxicity when compared to normal tissues [229], which make them more vulnerable to high temperatures thereby enabling PTT to destroy tumor cells selectively while protecting healthy cells around the tumor area [230]. This facilitates PTT administration along with other synergistic therapeutic strategies.

Single-walled carbon nanotubes containing polyvinylpyrrolidone and phosphatidylcholine were functionalized for delivery of curcumin which showed augmented curcumin delivery into cancerous cells within 4 h [231]. This formulation showed enhanced uptake of curcumin up to 6-fold greater than native curcumin and increased the blood concentration of curcumin by as high as 18-fold. This photothermal ablation effect was further demonstrated in in vivo models wherein it led to a reduction in the tumor volume and weight.

Phytochemical compounds have been used along with magnetic nanoparticles towards phototherapy as well as drug delivery [232]. Iron oxide nanoparticles capped with eugenate (4-allyl-2-methoxyphenolate) were generated via green synthesis using *Pimenta dioica*, a medicinal plant [233]. This formulation demonstrated favorable biocompatibility in human embryonic kidney 293 cell line (HEK293) and the human cervical cancer (HeLa) cell line in addition to showing robust efficacy of hyperthermia production upon irradiation with laser at near infra-red (NIR) wavelength. Curcumin was loaded within Fe_3_O_4_ magnetic nanoparticles coated with silica towards generating singlet oxygen and hyperthermia [234]. Combined treatment with this formulation and PDT resulted in reduction in tumor volume by 58%, while a combination of these nanoparticles with PDT and PTT showed 80% reduction in tumor volume, which was attributed to the synergistic effects obtained by ROS production and hyperthermia at the tumor site [234]. EGCG, when combined with PDT, augments anticancer effects in vitro and in vivo [235].

Selective targeting of cancer cells occurs through various mechanisms. Passive targeting utilizes the unique characteristics of tumor vasculature, which is typically more permeable and has defective lymphatic drainage compared to normal tissues [236]. This phenomenon, known as the Enhanced Permeability and Retention (EPR) effect, allows for large macromolecular compounds (over 40 kDa) nanoparticles to preferentially accumulate in tumor tissues [237]. This process enables the targeted delivery and retention of these compounds within solid tumor tissue, making it an essential mechanism for effective cancer treatment. Another approach is active targeting, that involves the presence of specific markers on the surface of cancer cells that facilitate the preferential uptake of photosensitizers (PS) by cancer cells compared to healthy cells. Additionally, the tumor microenvironment, characterized by factors such as hypoxia and acidity, can enhance the sensitivity of cancer cells to ROS-induced damage, leading to selective destruction of cancer cells while sparing healthy cells. These combined factors contribute to the effectiveness of phototherapy in targeting and eliminating cancer cells while minimizing harm to surrounding healthy tissues [212].

## 8. Negative Effects of Phytochemicals

Certain phytochemicals have been found to be toxic to humans and these phytotoxins have been shown to act as anticholinergic, adverse gastrointestinal irritants, cyanogens, cardiac glycosides, stimulants of the central nervous system, and hallucinogens [238]. Some polyphenols have been associated with genotoxic/carcinogenic effects and were shown interfere with the biosynthesis of thyroid hormone [239].

Neuropathy, specifically peripheral and autonomic sensory-motor neuropathy is a dose-circumscribing and dose-dependent negative effect often demonstrated in cancer patients undergoing vincristine treatment [240]. Neuropathy could be classified as chronic or acute and typically emerges within two weeks of initiation of treatment. The major common type, Vincristine-induced peripheral neuropathy (VIPN) frequently manifests as muscle weakness, paresthesia, areflexia, neurotic pain, wrist, and foot drop [241].

Although several flavonoids can affect the normal function of the thyroid gland, phytoestrogens are the major substances of concern interfering with thyroid metabolism and function. Several studies have shown that quercetin and phytoestrogens could induce thyroid disruption [242]. Their toxic effects include mutation and carcinogenicity, kidney and liver toxicity, negative effects on thyroid and reproductive functioning, and elicitation of disorders in intestinal flora. The toxicity mechanism is complex, and currently available evidence shows that naturally occurring flavonoid glycosides act on various targets at different doses in vitro and in vivo. Although most categories of flavonoids have been deemed safe, flavonoids recommended as food supplements must be assessed for tolerable maximal intake level due to reports of flavonoid toxicity.

Capsaicin functions as a co-carcinogen in 12-O-tetradecanoylphorbol-13-acetate (TPA)-promoted carcinogenesis of the skin in vivo and therefore caution must be exercised when administering capsaicin for topical application on a prolonged basis, particularly in the presence of cancer promoters, including solar UV radiation exposure [243].

Cycasin and its metabolite, methylazoxymethanol (MAM), are usually extracted from roots and seeds of cycad plants [244]. These plants are potently poisonous, and the toxicity due to ingestion of seeds is primarily caused by its misuse as a food source, as an agent to augment health, for precluding cancer, for cosmetic purposes, and for the treatment of gastrointestinal disorders. MAM is considered a genotoxic metabolite and has been shown to be involved in targeting of cellular processes associated with cancer development and neurodegeneration [245].

Genistein is an isoflavone phytoestrogen occurring in soybeans, fava beans, and red clover. Another principal category of phytoestrogens are lignans of which matairesinol is found in several foods such as fruits, vegetables, whole grains, and oil seeds [246]. Dietary phytoestrogens have been shown to contribute to the development of colorectal cancer in women and prostate cancer in men [247]. Localized production of estrogen is catalyzed by the enzyme aromatase, which is differentially regulated in healthy and cancerous breast tissue. Soy supplements used to ameliorate menopausal symptoms have been shown to elicit the growth of MCF-7 breast cancer cells by increasing aromatase biosynthesis and activity associated with breast cancer [248]. Particularly, genistein has been shown to obstruct the inhibitory activities of aromatase inhibitors, including letrozole [249] and fadrozole [248] against growth of MCF-7 breast cancer cells in a xenograft model and in vitro, respectively. Hence, women under treatment with aromatase inhibitor must be cautioned against the consumption of soy products.

Aristolochic acids constitute a category of compounds used in traditional herbal remedies since ancient times. They are known for their anticancer effects [250]. However, many studies have shown that aristolochic acid exposure leads to high occurrence of cancer involving the urinary tract and kidney [251].

Tetracyclic diterpenoids belong to the phorbol ester category of compounds known for their robust tumor-promoting effects. The seed-derived oil of the Croton plant has been used in several herbal medicines for years and contains Phorbol 12-myristate 13-acetate, which reportedly increases neutrophil and white blood cell counts in patients harboring solid tumors [252]. It also interferes with the migration and proliferation of thyroid cancer cells [253] and inhibits the growth of prostate cancer cells when used in combination with the anticancer drug, paclitaxel [254]. Nevertheless, it is also known to potently promote skin cancer [255]. Therefore, if this compound is used appropriately, it can treat lymphomas and leukemia despite their potential to induce skin cancer.

Pyrrolizidine alkaloids (PAs) riddelliine or comfrey occur in teas and are likely the most used among herbs in the present times. PAs have been reported to elicit liver cancer [256] in animal models. Dehydro-PAs interact with DNA and cellular proteins, causing cancer and genotoxicity. Particularly, they cause skin cancer by generating ROS, leading to lipid peroxidation [257].

Taken together, consumption of some phytochemicals can induce carcinogenesis, and the internet affords a huge marketplace for such kinds of products. Clinically notable adverse reactions to several unconventional remedies obtained via the internet have been observed [258,259]. Therefore, consumers need to be conscious that dietary supplements consisting of phytochemicals and related compounds are practically unregulated, due to which manufacturers of such products are not required to demonstrate the health benefits and safety of these products before their release into the market.

## 9. Expression of Monoclonal Antibodies (MABs) in Plants

There are other ways that plants can be a preferable solution to the world’s most pressing problems. Low cost, high scalability, and low risk of human pathogen contamination are the hallmarks of plant-based systems for producing MABs [260]. Glycoengineering of plant hosts offers another advantage over mammalian cell-based systems to produce MABs. Owing to their pluripotency, plants have the capability for regeneration from somatic cells [261]. Plants have been used for both transient and stable expression [260,262].

Transgenic plants are the most appropriate plant-based system for MAB production on a large scale.

MABs have also been generated by transient expression using recombinant plant viruses and agroinfiltration. Plant virus vectors can be employed for transient MAB expression more speedily than that of transgenic plants. Further, TMV vectors harboring light and heavy chains have been used to express full-sized MABs in *N. benthamiana* [263,264]. The expression platform commercially known as ‘MagnICON’ proved to be very effective for the high-yield production of MABs in *N. benthamiana*. A successful example of the exploitation of this technology is the Phase 1 clinical study conducted on chimeric antibodies for the treatment of B cell follicular lymphoma essentially by using these molecules as individualized idiotype vaccines created from patient’s own cancer cells [265]. The rapid and high-level expression offered by plants permitted to proceed from biopsy to the individualized vaccine in less than 12 weeks [266].

Additionally, other plant-based systems, such as hairy roots cultures, plant tissue cultures, plant cell suspension cultures, and aquatic plants, can be used similar to mammalian cell cultures. The candidate MABs and other heterologous proteins could be secreted into the prevailing culture medium, which enables facile harvest and purification.

## 10. Recent Developments Involving the Expression of Plant-Based Monoclonal Antibodies Against Cancer

Bulaon et al., 2024, designed and produced a plant-based bispecific monoclonal antibody (bsAb) capable of recognizing both cytotoxic T-lymphocyte-associated protein 4 and programmed cell death ligand 1 within a single molecule called dual variable domain immunoglobulin atezolizumab × 2C8 [267]. This bsAb was expressed transiently in *N. benthamiana*, wherein it demonstrated capability to bind cytotoxic T-lymphocyte-associated protein 4 and programmed cell death ligand 1 proteins in vitro. This antibody significantly blocked tumor growth in murine models harboring CT26 colorectal tumor while being tolerable and safe. Table 4 shows some examples of the recently generated MABs in plant systems.

For many cancer malignancies, Durvalumab (called Imfinzi) that targets PD-L1 is currently being employed for immunotherapy. This IgG1 antibody Fc region has been genetically engineered to decrease FccR interactions towards enhancing the inhibition of interactions between PD-1 and PD-L1 without depleting immune cells expressing PD-L1 [273]. *N. bethanmiana* was engineered to express four Durvalumab variants, namely the wild-type IgG1 and LALAPG, its ‘Fc-effector-silent’ variant harboring modifications to enhance antibody half-life, as well as IgG4S228P and PVA, its variant having Fc mutations to reduce FccRI binding. Additionally, Durvalumab variants were generated in their afucosylated form and their decorated form with 1,6-core fucose [273]. Plant-based durvalumab variants interact with recombinant PD-L1 as well as gastrointestinal cancer cell PD-L1 to effectively inhibit their binding with PD-1 on T cells, resulting in enhancement of their activation. Moreover, those plant-derived antibody variants harboring core fucosylation and Fc amino acid mutations show positive impacts on their therapeutic potential. In comparison with Imfinzi, DL-IgG4 (PVA) S228P and DL-IgG1 (LALAPG) exhibit lesser affinity for the CD32B inhibitory receptor that can be therapeutically advantageous. Significantly, DL-IgG1 (LALAPG) demonstrated augmented FcRn binding, a vital determinant of IgG serum half-life [273].

Conventional MABs like Trastuzumab face shortcomings during treatment of Human Epidermal Growth Factor Receptor 2 (HER2)-positive breast cancer, specifically in patients who develop drug resistance. Park et al., 2024, report a study wherein they express plant-produced anti-HER2 variable fragments of a camelid heavy-chain domain (VHH) fragment crystallizable region (Fc) KEDL(K) antibody, showing a capability to function as a potent alternate treatment to overcome such limitations [277]. This plant-derived antibody proved to have specifically high affinity for breast cancer cells that are HER2-positive, inclusive of those that are resistant to Trastuzumab. Further, in mice that are immune-deficient, this plant-based anti-HER2 VHH-FcK antibody shows superior anticancer activity, particularly against Trastuzumab-resistant tumors, underscoring its potential as effective immunotherapy for HER2-positive breast tumors that are Trastuzumab-resistant.

Jin et al., 2023, expressed anti-human epidermal growth factor receptor 2 (HER2) VHH-FcK MABs and anti-colorectal cancer large single chain (LSC) CO17-1AK through cross-pollination of plants expressing anti-HER2 VHH-FcK and LSC CO17-1AK, respectively, both of which targeted proteins in SKBR-3 human breast and SW620 human colorectal cancer cell lines correspondingly and inhibited cell migration to levels equivalent to that of their respective parental antibodies [270].

Bulaon et al., 2023, report the production of an anti-CTLA-4 antibody, 2C8, by rapid transient expression in *N. benthamiana* plants [278]. This anti-CTLA-4 2C8 MAB interacts with murine and human CTLA-4 proteins with comparable efficiency to that of one of the Fcg receptors. Additionally, it demonstrated equivalent antitumor efficacy to that of the commercially available anti-CTLA-4 MAB (Ipilimumab Yervoy^®^) in a humanized murine tumor model, implying that the plant-derived anti-CTLA-4 MAB has similar therapeutic potential to that of the clinically efficient Ipilimumab. Considering that Ipilimumab is expensive and unaffordable in the developing world, plant-based production of anti-CTLA-4 2C8 MAB is more appealing for rapid expression, facile scale-up, and economical manufacture of such recombinant therapeutics.

Immune checkpoint inhibitors (ICIs) are a category of immunotherapeutic agents with the capability to alleviate the immunosuppressive environment exerted by neoplastic cells. Tumorigenic cells use one of the most universal checkpoints, the programmed cell death protein 1 (PD-1)/programmed death-ligand 1 (PD-L1), for evading the immune system by eliciting apoptosis and blocking cytokine production and proliferation of T lymphocytes. Presently, the most commonly used ICIs that target the PD-1/PD-L1 checkpoint are MABs nivolumab and pembrolizumab, which interact with PD-1 occurring on T-lymphocytes and block the binding with PD-L1 expressed on tumorigenic cells.

Nivolumab and pembrolizumab are the most commonly used ICIs for treatment of several cancers, such as Hodgkin lymphoma, melanoma, lung, breast, and colorectal cancers [279,280,281]. These two antibodies block the PD-1/PD-L1 immune checkpoint, resulting in CTL activation and apoptosis induction in tumorigenic cells via T-cell-enabled cytotoxicity [282]. Both of these ICIs significantly augment the rates of survival of patients having a diverse range of cancer types. Nevertheless, the cost of these therapies currently in the market is exorbitantly high and therefore inaccessible to patients in developing countries [283], primarily owing to the expensive mammalian cell platform used for their expression. Hence, plant-based molecular farming of these antibodies is appealing due to its potential to greatly decrease the capital required for the manufacture of these ICIs [271,280]. Nivolumab and pembrolizumab have been transiently expressed in *N. benthamiana* leaves at yields as high as 140 mg/kg FLW and 340 mg/kg FLW, respectively, which amounted to USD 4200 and 18,000 worth of these two antibodies correspondingly in 1 kg of leaves [275].

The first antibody that targeted the immune checkpoint PD-L1 was Atezolizumab (Tecentriq), which is currently one of the most widely used drugs in anticancer therapy. Nevertheless, this anti-PD-L1 antibody is expressed in mammalian cells that incur high costs for manufacture, circumscribing the access of antibody treatment for cancer patients. Plant-based Atezolizumab upon transient expression in *N. benthamiana* showed high levels of expression within 4–6 days following infiltration. Purified plant-expressed Atezolizumab had no glycosylation and was able to bind to PD-L1 with equivalent affinity to that of Tecentriq [271]. Additionally, this plant-produced antibody inhibited tumor growth with an efficacy comparable to that of Tecentriq in murine models. This substantiates the capability of plants to serve as efficacious platforms for production of immunotherapeutic antibodies, and therefore plants can be used to mitigate the cost of currently used anticancer drugs [271].

Varlilumab, a CD27-targeting monoclonal antibody, in its recombinant form was expressed in leaves of *N. benthamiana* within as little as 8 days of infiltration and was shown to assemble successfully [276]. The RNA silencing suppressor, p19 was co-expressed along with Varlilumab, which resulted in the accumulation of the Mab averaging to about 174 µg/g of the fresh leaf weight. This was greater when compared to the yield of nivolumab (140 µg/g of fresh leaf weight) produced in transient geminiviral expression system in *N. benthamiana* [281]. Purified plant-produced Varlilumab assembled properly into a tetramer and showed in vitro efficacy comparable to that of the commercial Varlilumab expressed in mammalian cells.

## 11. Plant-Based VNPs and VLPs Against Cancer

Plant viruses are a group of pathogens that infect plants. They contain a protein cage (capsid) and nucleic acid (RNA or DNA) [284,285]. The capsid naturally encloses the nucleic acid via a straightforward supramolecular self-assembly mechanism from many copies of one or a few different types of coat protein (CP) subunits [285]. CP subunits without the nucleic acid can assemble as virus-like particles (VLPs) under in vivo or in vitro conditions [286]. Thus, two key nanostructures based on plant viruses, as plant virus nanoparticles (PVNPs), are available: the complete plant virus virion or virus nanoparticles (VNPs) and VLPs. PVNPs can be naturally isolated from plants or produced by ‘molecular farming [287]. PVNPs have various morphologies, including icosahedral, road like, bacillus, and filamentous. PVNP’s noninfectious nature in animals, biocompatibility, biodegradability, and non-teratogenic properties significantly reduce their in vivo toxicity [288]. PVNPs can modify via genetically and physicochemical strategies to produce modified PVNPs [285]. Genetic modification involves incorporating non-native biological elements (specific amino acids, peptides, tags, and proteins) into the CP of PVNPs, or removal of residues from CP [289]. Genetic modification can be performed for displaying specific ligands or further modifications [289]. Chemical modification techniques depend on the utilization of activated conjugates to react with functional groups of natural or unnatural amino acids that are found on the capsid, within specific solution conditions of pH and salt concentration. Another method for manipulating is open pore structures on PVNPs. Pores are responsive to environmental stimuli, and undergoing conformational changes, allowing controlled molecule entrapment [289]. For instance, red clover necrotic mosaic virus (RCNMV) encapsulates cargo through infusion triggered by stimulus-induced capsid pores opening and closing [290]. Self-assembly is an effective method for engineering of PVNPs, where purified CPs assemble around a specific cargo [284]. The extent of PVNP modification depends on charge, shape, size, shielding, and targeting, which are all influenced by various factors, such as receptors or environmental factors [289]. The potential of PVNP-based modifications exhibits diverse yet distinct dimensions and configurations, making them highly customizable in terms of size and shape and capable of loading various natural and synthetic payloads [285,289,291]. In the context of cancer, modification of PVNPs can be applied to create and improve a variety of therapies, including chemotherapy, gene therapy, immunotherapy, and vaccines (Table 5).

## 12. PVNPs as Delivery Nanosystem in Cancer

Delivery nanosystems are widely used to improve the safety and efficacy of encapsulated therapeutic/imaging agents [300,301]. Recently, PVNPs have been utilized as an innovative platform for delivery applications [289]. They can address therapeutic/imaging agent’s limitations by encapsulating them on their interior/exterior surfaces, and enhancing their effectiveness and safety [285,289]. Structurally, untargeted PVNP formulations have the potential to exhibit enhancement in their concentration within tumors via passive-targeting delivery, utilizing the “enhanced permeability and retention (EPR) effect [285]. However, the most important challenge of non-targeted PVNPs is clearance by phagocytes, even when modified with polyethylene glycol (PEG), due to the existence of anti-PEG antibodies. To address this concern, a potential strategy involves the conjugation of serum albumin (SA) to PVNP-based nanocarriers. An investigation carried out on Balb/C mice revealed that SA-conjugated TMV, which was “camouflaged” using SA, exhibited decreased antibody recognition and enhanced pharmacokinetics. Furthermore, the attachment of specific ligands to the PVNP surface allows for the targeted delivery of PVNP formulations to target sites. Such ligands include GE11 (short peptide comprising 12 amino acids) [302], folate [303], Herclon (a monoclonal antibody (MAB)), peptide F3, epidermal growth factor-like domain 7 (EGFL7) [304], and CooP (a 9 aa peptide, sequence: CGLSGLGVA) [292].

Drug delivery: Anticancer therapeutic agents face numerous challenges, including high resistance, frequent recurrence, rapid drug elimination, and non-targeted distribution, leading to toxicity and limited clinical effectiveness. The use of targeted delivery systems and dose reduction have the potential to enhance therapeutic outcomes. Recently, it was shown that PVNPs can be ideal for loading and delivery of anticancer agents (e.g., nucleic acids, peptides, proteins, small molecules, and nanoparticles). They protect encapsulated payloads from degradation, facilitate their targeted delivery, selective cytotoxicity, and sustained efficacy even at very low doses. Previous work has shown that PVNPs can encapsulate small molecule therapeutics (DOX, MTO, phenanthriplatin, gemcitabine, and cisPt), enhancing their accumulation in tumor tissues and causing cytotoxic effects (Figure 3A) [305,306,307,308]. For example, the loading of DOX utilizing CPMV, RCNMV, Johnson grass chlorotic stripe mosaic virus (JgCSMV), or the prodrug DOX via Physalis mottle virus (PhMV)-based VLPs has demonstrated significantly improved antitumor activity both in vitro and in vivo [309,310,311]. Notably, the DOX-loaded PVNPs elicited HCC cell death at IC50 values of 10–15 nM, which is 20 times more effective than free DOX. Studies find that the TMV can transport cisplatin and mitoxantrone, demonstrating exceptional cytotoxic effects, making it a promising candidate for reducing anticancer medication dosages (Figure 3A) [306,312].

Additionally, CPMV was utilized for the targeted delivery of mitoxantrone to treat glioblastomas, resulting in increased cytotoxicity against glioma cells [316]. TMV has been used to attach valine-citrulline monomethyl auristatin E, an antimitotic drug, to target non-Hodgkin’s lymphoma. The TMV formulations showed potent cytotoxic effects against non-Hodgkin’s lymphoma cell lines in vitro, with an IC50 value of around 250 nM. This targeted delivery capability was used to enhance the potency and effectiveness of phenanthriplatin, a cationic mono-functional DNA-binding Pt(II) anticancer drug, resulting in a significant reduction in tumor size [307,317,318]. Filamentous plant virus nanoparticles have shown potential for displaying protein drug delivery in cancer therapy. For example, PVX nanocarriers-based TRAIL/Herceptin delivery has shown enhanced therapeutic efficacy in cancer treatment through targeting specific receptors. The use of PVX nanocarriers for Herceptin delivery in breast cancer treatment has shown enhanced therapeutic efficacy, as it effectively induces apoptosis in HER2 positive cell lines, demonstrating the potential of this innovative approach [319]. Potato virus X has been used as a nanocarrier to deliver TRAIL, a protein drug that induces apoptosis in cancerous cells. This method reduced tumor growth in mouse models of human triple-negative breast cancer (Figure 3B) [313]. Le et al. developed a nanocarrier using potato virus X for targeted doxorubicin delivery, showing efficacy against ovarian, breast, and cervical cancer cell lines, inhibiting tumor growth in mouse models [305,320]. Recently, TBSV-based PVNPs have shown potential in drug delivery to Sonic hedgehog medulloblastoma. Targeted TBSV with CooP (a 9 aa peptide, sequence: CGLSGLGVA) (TBSV-CooP), selectively transports DOX to medulloblastoma, enhancing cell proliferation and apoptosis [292].

The bioconjugation process, involving the linking of epidermal growth factor-like domain 7 (EGFL7) (a protein predominantly expressed in endothelial cells) to CPMV, yielded a modified protein capable of targeting tumor-associated neovasculature with a remarkable level of precision [304]. Recently, a peptide-guided tomato bushy stunt virus (TBSV)-based nanocarrier system loaded with DOX has been utilized for cell-specific delivery. Marchetti et al. (2023) harnessed TBSV-based VNPs with CooP peptide, a homing peptide designed for medulloblastoma tumors. Encapsulating DOX within TBSV-CooP enhanced cell death and proliferation, underscoring their effectiveness in targeting brain tumors. The internalization of the TBSV-based nanocarrier platform, equipped with the C-terminal C-end rule (CendR) peptide RPARPAR (RPAR) (TBSV-RPAR), loaded with DOX, displayed selective cytotoxicity towards cells expressing the receptor neuropilin-1 (NRP-1) [321]. Another research effort showcases the utilization of iRGD peptides to target tumor neovasculature on PhMV-like nanoparticles, resulting in rapid uptake and heightened tumor homing. This strategy presents a promising avenue for delivering targeted molecular cargo to tumors [322].

PVNP can act as PTT and PDT agents in the treatment of tumors. PVNP-based PTT/PDT agents, which absorb photons, have the capability to produce heat or ROS to eliminate cancer cells [323,324]. For example, the conjugation of a photothermal biopolymer polydopamine (PDA) on TMV followed by exposure to near-infrared laser in combination with immunotherapy and multimodal magnetic resonance/photoacoustic imaging presents a promising strategy and theranostic approach for cancer models in vivo [325]. Incorporating the porphyrin-based photosensitizer drug, Zn-Por, into TMV and tobacco mild green mosaic virus (TMGMV), has been proven to lead to a notable enhancement in the effectiveness of cell destruction. Notably, a five-fold increase in efficacy was observed in comparison to the free drug [324]. Researchers have developed a drug delivery approach using TMV protein nanotubes as a carrier for ovarian cancer treatment. The nanochannel was loaded with cisplatin, achieving a loading efficiency of 2700 cisPt^2+^ per TMV. TMV-cisPt showed superior efficacy against ovarian tumor cells, reducing tumor burden and increasing survival in mouse models [326].

Gene delivery: Nucleic acid therapeutics are widely used for treating diseases, but the main challenge is the safe and effective delivery of nucleic acids. Initially, mammalian viruses were preferred, but concerns about immunogenicity and integration led to alternative delivery methods like PVNPs, which combine viral and nonviral characteristics [327]. PVNPs have the capability to enhance suboptimal cellular uptake, instability caused by nucleases, and inefficiencies in the delivery of nucleic acids by encapsulating various types of RNA, such as heterologous RNA, siRNAs, mRNA, and CpG-ODNs [328]. Lam et al.’s research marked a milestone in PVNP-mediated genetic therapy, showing that an icosahedral PVNP can target and deliver a siRNAs targeting green fluorescent protein (GFP) or Forkhead box protein A1 (FOXA1). This study shows that siRNA molecules can be loaded into cowpea chlorotic mottle virus (CCMV) nanoparticles, and only CCMV with appended cell penetrating peptide (CPPs), such as M-lycotoxin peptide L17E, was effective in silencing the FOXA1 gene [329]. It has been demonstrated that VLPs with CCMV capsid along with mRNA-EGFP as cargo and reporter gene, have the ability to directly transfect eukaryotic cell lines without adjuvants and deliver nucleic acid for translation [330]. CP-miR-26a (CP26a) VLPs were created by self-assembling purified CP from CCMV, retaining its structure and protecting miR-26a from digestion. CP26a showed similar cellular uptake efficiency, osteogenesis promotion ability, and better biocompatibility compared to Lipofectamine2000-miR-26a [331]. An illustration of this is seen with brome mosaic virus (BMV) and CCMV, which can be utilized to carry the siRNA Akt1 (siAkt1) for uptake by tumor cells [332]. Furthermore, CCMV, in combination with siRNAs designed to target FOXA1, a crucial transcription factor in the forkhead box (FOX) protein family, has been shown to induce gene silencing in the MCF-7 breast cancer cell line [329]. CCMV can be used for gene delivery, with a virus-like particle (VLP) encapsulating anti-miR-181a oligonucleotides to knock down ovarian cancer cells (Figure 3C). This study showed higher knockdown efficacy and reduced cancer cell invasiveness, highlighting the potential of plant-derived VLPs as nucleic acid carriers [314].

## 13. PVNPs as Imaging Agents

Nanoengineering of PVNPs presents a myriad of possibilities for the loading and modification of contrast agents [333]. Guanidinium agents, referred to as PVNP-based dyes, are commonly employed in preclinical diagnostic imaging. These contrast agents, based on PVNPs, hold promise for the development of features such as prolonged circulation, specific targeting capabilities, and efficient delivery to tumors in vivo. An example of this is the utilization of PhMV-like nanoparticles loaded with the fluorescent dye Cy5.5 and paramagnetic Gd(III) complexes, with PEGylated particles being linked to targeting peptides to monitor a human prostate tumor model using near-infrared fluorescence and magnetic resonance imaging (Figure 3D) [315]. PVNPs can also be adorned with bombesin peptides, PEG, and near-infrared fluorescent dyes [334]. Through the loading of Dy^3+^ and Cy7.5 into TMV nanoparticles and their conjugation with a Dy^3+^ dye and near-infrared fluorescence (NIRF) dye, significant transverse relaxation of targeted PC-3 prostate cancer cells and tumors was successfully achieved in vitro and in vivo under ultra-high-strength magnetic fields [335].

CPMV loaded with near-infrared dye (Alexa Fluor 647) and PEG, in addition to conjugation with the pan-bombesin analog, [β-Ala11, Phe13, Nle14] bombesin-(7–14), has the ability to specifically target the gastrin-releasing peptide receptor, which is known to be highly expressed in human prostate cancers. The process of tumor homing was observed by utilizing human prostate tumor xenografts on the chicken chorioallantoic membrane model, employing intravital imaging techniques [336]. The elongated PVX can be genetically modified to exhibit either green fluorescent protein (GFP) or mCherry as markers for optical imaging in human cancer cells and in a preclinical mouse model [337]. Plant viruses, particularly TMV, can serve as a foundation for contrast agents in magnetic resonance imaging (MRI); TMV particles can be filled with Gd(DOTA) within the internal channel of TMV and the surface covered with silica, thus enhancing T1 relaxivities in comparison to uncoated Gd-loaded TMV [338]. The display of GE11 on PVX and the attachment of PVX-GE11 filaments with fluorescent markers can be precisely aimed at the epidermal growth factor receptor (EGFR). The detection and visualization of cells were illustrated using cell lines of colorectal adenocarcinoma, human skin epidermoid carcinoma, and triple-negative breast cancer (A-431, HT-29, MDA-MB-231), all of which displayed varying levels of EGFR upregulation [302].

## 14. PVNPs as Theranostic Agents

The distinctive structural and chemical characteristics of PVNPs render them highly appropriate for the integration of therapeutic and diagnostic agents’ capabilities for in vivo applications [339]. Metal–phenolic networks (MPNs) derived from plant viruses like TMV, PVX, and CPMV have demonstrated advantageous optical properties, cytocompatibility, and remarkable cell-destructive performance during photothermal therapy, particularly when loaded with complexes of tannic acid (TA), metal ions (e.g., Fe^3+^, Zr^4+^, or Gd^3+^), or fluorescent dyes (e.g., rhodamine 6G and thiazole orange) and exposed to 808 nm irradiation [340]. Gd-loaded TMV particles coated with polydopamine (PDA) inspired by mussels represent biocompatible nanotheranostic agents that facilitate multimodal imaging and photothermal therapy (PTT) in PC-3 prostate cancer cells [341]. The utilization of SA-coated TMV laden with chelated gadolinium (DOTA) for detection via magnetic resonance imaging and the loading of DOX could enable the monitoring of disease progression, thus offering insights into the effectiveness of the drug delivery approach [341]. Engineered TMV-MOF (metal–organic framework) hybrid nanoparticles enhanced the retention of these VNPs in murine models [342]. By coating the TMV encapsulated with Cy5 with zeolitic imidazolate framework-8, particles of Cy5-TMV@ZIF were created, resulting in a 2.5 times higher fluorescence retention time compared to Cy5-TMV alone. These Cy5-TMV@ZIF particles exhibited resistance to harsh conditions, were non-toxic, and displayed high stability [342]. Tobacco mosaic virus (TMV) particles impregnated with a metal-free paramagnetic nitroxide organic radical contrast agent (ORCA) were developed as probes for electron paramagnetic resonance and magnetic resonance imaging to detect superoxide. These probes demonstrated enhanced in vitro r1 and r2 relaxivities, acting as both T1 and T2 contrast agents, highlighting their potential for preclinical and clinical MRI scanning [343]. In a separate study, TMV VNPs were engineered to target VCAM-1, the vascular cell adhesion molecule, and loaded with Gd-dodecane tetraacetic acid (GdDOTA). This led to highly sensitive identification and visualization of atherosclerotic plaques in ApoE-/- mice using minimal contrast agent doses, resulting in improved relaxivity and moderate tumbling of the Gd-DOTA-TMV carrier with enhanced signal-to-noise ratio. Furthermore, these conjugates exhibited heightened imaging sensitivity, leading to a 40-fold reduction in Gd dosage compared to standard clinical doses [344].

## 15. PVNPs as Vaccine and Immunotherapy Agents

Recent studies suggest that intratumoral immunotherapy (IT-IT) of certain PVNPs can induce anti-tumor immune responses within the tumor microenvironment [345,346]. PVNPs’ immune stimulation against tumors is due to non-self [328] recognition. They can be identified by innate immune cells’ pattern recognition receptors (PRRs), specifically Toll-like receptors (TLRs) [328]. PVNPs, upon interaction with surface or endosomal TLRs on antigen-presenting cells (APCs), trigger the secretion of cytokines, chemokines, and interferons, recruiting and activating anti-tumor immune cells [328]. Studies have shown that PVNP’s immunostimulatory effects can be mediated by nucleic acid content and multimeric coat protein assemblies [328]. One example is CPMV, wherein TLR2 and TLR4 identify capsid or empty CPMV; TLR-7/8 are responsible for recognizing positive-strand RNA genomes (Figure 4A) [328,347,348,349].

Recent studies show that CPMV with two different RNA genomes packaged separately into identical protein capsids can activate innate immune cells, inducing pro-inflammatory cytokines and suppressing immunosuppressive cytokines, with comparable efficacy [347]. The use of intratumoral immunotherapy (IT-IT) of eCPMV in canine inflammatory mammary cancer (IMC) demonstrate notable rise in neutrophil populations, T and B lymphocytes within the tumor microenvironment, and induced the anti-tumor response [351]. Based on several findings, it is evident that IT-IT of PVNPs such as PVX [352], papaya mosaic virus (PapMV) [353], TMV [354] and CCMV, Sesbania mosaic virus (SeMV) [355], and alfalfa mosaic virus (AMV) [356] are increasingly recognized as highly effective in situ vaccination agents for various types of cancers. Recently, Zhao et al. have demonstrated that conjugation of targeted peptides of mannose receptor, namely CSPGAK (CD206s, 561.7 Da) and CSPGAKVRC (CD206, 920.1 Da) (CD206 and CD206s), to CPMV functions as a promising avenue for cancer immunotherapy directed at M2 macrophages [357].

While these PVNPs exhibit variability in their capacity to act as immune stimulants, they differ in additional characteristics, such as their potential to be modified for delivering cancer antigens [346]. Utilizing PVNP-based cancer vaccines enables the initiation of immune responses specific to tumor-associated antigens. Recent research has detailed the utilization of various PVNPs, including icosahedral CPMV, CCMV, SeMV, PhMV, and filamentous PVX in HER2-specific cancer vaccines (Figure 4B) [350,352,358]. CPMV has also been employed for the delivery of the immunogenic cancer-linked testis antigen NY-ESO-1 [294].

PVNPs can undergo modifications to load immunoadjuvants and improve antitumor efficacy. An illustration of this concept is the utilization of CCMV loaded with CpG oligonucleotides to stimulate the activation of tumor-associated macrophages (TAMs) both in vitro and in vivo (Figure 4C) [315,359,360]. The attachment of TLR3 or TLR7 agonists to various PVNPs such as CPMV and CCMV VNPs has been shown to boost immune cell activation and the generation of the pro-inflammatory cytokine interleukin 6. Specifically, the linking of the TLR7 agonist 2-methoxyethoxy-8-oxo-9-(4-carboxybenzyl) adenine (1V209) to CPMV and CCMV resulted in diminished tumor growth and enhanced survival rates in mice [297]. Likewise, the administration of a nucleic acid based TLR3 agonist, polyinosinic acid with polycytidylic acid (poly(I:C)), through CPMV led to an increase in survival in mice. These results emphasize the significance of combining and co-delivering TLR agonists to enhance their antitumor efficacy, with the multivalent presentation, prolonged presence in tumors, and precise targeting of innate immune cells by the PVNP carriers being pivotal factors in improving effectiveness. The results indicate that active microneedle, Pluronic F127, implantable polymeric hydrogels formulations effectively maintain the structure and function of CPMV for enabling slow-release immunotherapy for cancer [297,361,362].

## 16. PVNPs-Based Combination Therapies

Combining intratumoral PVNPs with other tumor therapies is being explored to improve treatment outcomes as demonstrated by the potent efficacy of chemotherapy, immune checkpoint therapy (ICT), radiation therapy (RT), along with PVNPs in mouse tumor models. Results suggest that utilizing CPMV particles in combination with RT can turn an immunologically “cold” tumor (with low number of tumor infiltrating lymphocytes (TILs)) into an immunologically “hot” tumor (with increase in TILs) [363]. ICT in cancer treatment has shown promising results but is limited to a minority of patients. Heightened efficacy of combined CPMV IIT and anti-Lymphocyte-activation gene-3 (LAG-3) treatment in a mouse model of melanoma have been observed wherein LAG-3 functions as a next-generation inhibitory immune checkpoint with broad expression across multiple immune cell subsets. Its expression increases on activated T cells and contributes to T cell exhaustion. LAG-3 is a novel inhibitory immune checkpoint that increases activated T cells and promotes T cell exhaustion [364]. Combination of CPMV and anti-4-1BB monoclonal antibody agonist is an effective dual therapy approach. Using murine models of metastatic colon carcinomatosis and intradermal melanoma, intratumorally administered CPMV + anti-4-1BB (CD137) dual provided a robust antitumor response, improved elimination of primary tumors, and reduced mortality compared to CPMV and anti-4-1BB monotherapies [365]. CPMV combination with anti-PD-1 peptides (SNTSESF) resulted in increased efficacy; however, increased potency against metastatic ovarian cancer was only observed when SNTSESF was conjugated to CPMV, and not added as a free peptide. This can be explained by the differences in the in vivo fates of the nanoparticle formulation vs. the free peptide; the larger nanoparticles are expected to exhibit prolonged tumor residence and favorable intratumoral distribution [366].

Furthermore, the coadministration of DOX via PVX + DOX enhanced the response of the PVX monotherapy through increased survival, which was also represented in the enhanced antitumor cytokine/chemokine profile stimulated by PVX + DOX when compared to PVX or DOX alone [367]. Combination therapy using CPMV and low doses of cyclophosphamide (CPA) has shown remarkable synergistic efficacy against 4T1 mouse tumors in vivo. The combination of CPMV and CPA increases the secretion of several cytokines, activates antigen-presenting cells, increases the abundance of tumor infiltrating T cells, and systematically reverses the immunosuppression. These results show that the combination of CPMV in situ vaccination with chemotherapy may become a potent new strategy for the treatment of tumors [368].

Photothermal therapy (PTT) is a promising treatment for cancer that targets tumors locally and enhances immune responses. Nevertheless, the efficacy of PTT when used alone is often limited systemically, prompting the exploration of combined treatment strategies. In pursuit of this objective, the TMV was utilized to deliver a small molecule immunomodulator, Toll-like receptor 7 agonist (1V209), while its surface was modified with the photothermal biopolymer polydopamine (PDA). The resulting complex of 1V209-loaded TMV coated with PDA was employed in the treatment of B16F10 dermal melanoma in C57BL/6 mice. This formulation, known as 1V209-TMV-PDA, was administered by intratumoral injection followed by irradiation with an 808 nm near-infrared laser. Notably, 60% of the mice treated with intratumoral 1V209-TMV-PDA and laser irradiation survived until the study endpoint, demonstrating a significant improvement compared to the 20% survival rate observed in the control group (Figure 4D) [325].The study aims to develop a vaccine for ovarian cancer using irradiated cancer cells (ICCs) as an antigen and cowpea mosaic virus (CPMV) adjuvants. Results show co-formulated CPMV–ICCs successfully withstood initial tumor challenges in mice, emphasizing the importance of simultaneous delivery [369].

## 17. Recent Developments of PVNPs Against Cancer

Nanoparticle delivery systems have the potential to improve pharmacokinetics, tissue targeting, and stability of encapsulated therapeutic and imaging agents [370]. Numerous scientific advancements have been achieved using bioinspired nanocarriers such as proteins, nucleic acids, and viruses. One of the natural nanocarriers are self-assembling viruses to effectively deliver cargos [371,372]. PVNPs are new candidates that can be used as promising nanoplatforms [370,373]. Currently, PhMV-based nanoparticles have shown potential as a nanocarrier platform for loading cargo into living organisms. However, their internal loading capacity is limited due to low reactivity. A structure-based approach has created mutants with enhanced reactivity towards thiol-reactive small molecules (e.g., doxorubicin and vcMMAE) and imaging agents (DOTA(Gd)), resulting in ten-fold increased reactivity towards chemotherapeutic and MRI agents [370]. A nanomedicine platform using CCMV-based nanoparticles modified with elastin-like peptides (ELPs) has been developed to address chemotherapy difficulties. The nanoparticles deliver DOX into tumor cells, stimulate immune responses, and hinder tumor growth in cancer models [374].

Protein-based vaccines from mammalian, bacteriophages, and plant viruses offer advantages in cancer immunotherapy due to their ability to modulate the immune system [375]. PVNPs, although non-infectious to mammals, are recognized by the immune system as strong adjuvants. Systemic administration of CPMV activates the innate immune system, aiding in the identification and destruction of cancer cells, leading to a lasting and adaptable immune response against metastatic cancers like colon, ovarian, melanoma, and breast cancer [376]. A study on 11 companion dogs with canine mammary cancer found that neoadjuvant intratumoral immunotherapy with empty eCPMV led to tumor reduction in both treated and untreated tumors. The reduction was observed across different stages, sizes, grades, and molecular subtypes. The injected tumors showed decreased DNA replication and increased dendritic cell activity, with increased levels of neutrophils, T lymphocytes, and plasma cells. The eCPMV immunotherapy was effective and had no negative side effects [296]. A cancer vaccine targeting S100A9, a major inflammation regulator, has been developed using plant virus and bacteriophage nanotechnologies. The vaccine significantly reduced S100A9 levels in tumor-bearing mice, protecting against lung metastasis. The vaccine also increased immunostimulatory cytokines and decreased immunosuppressive cytokines, potentially having wide-ranging implications in preventing metastasis due to its prevalence in multiple cancer types [377]. The study found that CPMV, a cytokine, effectively treated a metastatic ovarian tumor model without causing organ toxicity. The viral RNA persisted and could be detected two weeks after final administration. The study also showed that systemic administration of CPMV is safe and widely available. Recently, polymeric hydrogels with CPMV were surgically implanted into the peritoneal cavity for cancer immunotherapy. PVNPs derived from TuMV have been found to attract immunoglobulins G (IgG) due to gene fusion. A fluorescent nanoplatform was created to target cancer cells overexpressing EGFR, presenting a promising avenue for cancer theranostics [378]. Small TLR agonists of poly(I:C) are being developed as intratumoral immunotherapies, with rapid washout and poor immune cell uptake. To address these issues poly(I:C) is enclosed in nanoparticles made from CCMV. These particles enhance the function of macrophages and demonstrate effectiveness in reducing tumor growth and extending survival in mouse models of colon cancer and melanoma. When combined with oxaliplatin, CCMV-poly(I:C) shows even greater efficacy, significantly inhibiting tumor growth and improving survival rates [379]. PVNPs, when combined with immunotherapy, offer promising cancer treatment. In the future, PVNPs-based cancer vaccines could be integrated with traditional methods like radiotherapy and chemotherapy. In murine cancer models, combining CPMV’s immune cell activation with oxaliplatin’s immunogenic cell death led to increased median survival rates. This combination therapy effectively altered the tumor microenvironment, leading to enhanced tumor cell death. The research underscores the potential for combining chemotherapy with PVNP intratumoral immunotherapy in clinical settings, highlighting the potential of PVNP-based cancer vaccines in the future [299]. Cryoablation and intratumoral CPMV, whether used separately or together, showed strong effectiveness against treated HCC tumors. However, only the combination of cryoablation and CPMV was able to inhibit the growth of untreated tumors, indicating an abscopal effect [380]. Generally, advancements in nanobiotechnology are expected to improve PVNPs, leading to advancements in tumor diagnosis, treatment, and prevention. Recent tumor screening and antigen discovery have contributed to personalized immunotherapy. By using optimized nanoadjuvants based on PVNPs, along with immune-related molecules like cytokines or immunopotentiators, nanocancer vaccines with targeted, safe, and controlled release can be developed.

## 18. Advantages of the Use of Plants for Production of Anti-Cancer MABs, Plant Viral Nanoparticles and Phytochemicals Against Cancer

The major advantages of the use of phytochemicals are their capabilities to induce multiple signaling networks, that enable cells to have greater decoding and signal processing properties. They preclude the advancement of cancer by decreasing survival and growth signals of tumor cells. Further, diverse phytochemicals can control crosslinked biosynthetic pathways such as the NF-κB axis, glycolytic enzymes, DLC1 pathway, MAPKs and ROS driving multiple processes against the tumor. Phytochemicals by virtue of their effectiveness on complex signaling cascades, inhibit cancer cell evasion caused by single pathways.

Plant-produced antibodies could be full-sized, single-chain antibody fragments, membrane-anchored scFv, bispecific scFv fragments, Fab fragments and chimeric antibodies. Unlike mammalian cells, recombinant plants can efficiently express secretory IgA. Plants are available on a widespread basis, grow rapidly and commonly mature following one growth season. Also, it is facile to bring the plant-based MABs to the market within brief timeframes and at large scales, which reduces the production cost. Further, plants decrease costs for screening for viruses, prions and bacterial toxins as they do not introduce human pathogens in contrast to production platforms such as transgenic animals or mammalian cells [381]. Furthermore, plants possess an endomembrane system as well as secretory pathway comparable with human cells, that are distinct from prokaryotic systems and bacteria. Whereas animal-based Abs elicit immune responses involving foreign or non-self-agents, plant-based Abs discount with such reactions. Similar to animal cells, they possess posttranslational modification mechanisms that enable them to be considered as biofactories for production of therapeutic Abs [382]. Plants that are glycoengineered have substantially greater degree of glycan homogeneity. Consequently, plants provide a powerful expression system for anticancer MABs production.

Plant viruses are appropriate for vaccine production due to their capability to be recognized as foreign molecules by the innate immune system via pathogen-associated molecular pattern (PAMP) receptors [383].They are not pathogenic to humans and elicit humoral antibody response as well as cell-mediated immune response [384,385] upon delivery via parenteral [386] and mucosal [387] routes. The genomes of plant viruses have been genetically engineered to enable the expression of foreign open reading frames.

VNPs can be used as novel nanomaterials having several favorable characteristics [388]. They self-assemble; are monodisperse, polyvalent, highly potent, and dynamic; provide structural uniformity; and can be produced within brief time periods. They are superior when compared with synthetic nanomaterials, as they are biodegradable and biocompatible. Ligands, including proteins, peptides, small chemical modifiers, and even other nanoparticles, can be encapsulated within the VNPs by self-assembly using a diverse range of bioconjugation chemistries, such as genetic engineering, chemical bioconjugation, mineralization, and encapsulation [389,390]. VLPs serve as robust vaccine candidates, as they mimic native virus conformations and induce intrinsic immunogenicity without dispensing with their inherent safety, which they accomplish by virtue of being bereft of the viral genome and hence being incapable of replication. Consequently, VLPs are increasingly favored as subunit vaccines, as they easily undergo internalization by the antigen presenting cells, leading to efficacious immune reactions. They are inimitable platforms for processing of antigens and presentation of epitopes to the immune system. Further, VLPs are popular for cancer immunotherapy due to their natural ability to elicit immune responses which prime the microenvironment of tumors and launch antitumor immunity. VLPs present themselves as multivalent, repetitive molecular scaffolds, as they are composed of viral capsid proteins in multitudinous copies that enable multivalent antigen presentation [391]. Vaccines based on VLPs provide superior immunogenicity in comparison to antigens occurring in their soluble forms. Also, they are characterized by their native adjuvant properties which dispense with the employment of further adjuvants to elicit potent immune responses.

Through genetic engineering, epitopes can be expressed on the plant virus capsids, resulting in a homogenous formulation while mitigating the heterogeneity of chemical conjugation methodologies [392]. Such vaccines could be successfully developed and purified from plant hosts via molecular farming, hence decreasing downstream processing and production costs [393,394]. Additionally, these plant virus-derived vaccines can be incorporated within polymeric implants or devices that can promote its shelf life while enabling extended antigen release [395].

## 19. Disadvantages of Plant-Based Platforms Against Cancer

The important negative effects of phytochemicals have already been discussed in Section 8. Additionally, many phytochemicals are associated with targeting of multiple signaling pathways shared between multiple cellular systems, which poses a challenge to the use of anticancer drugs based on phytochemicals.

Natural phytochemicals are ridden with drawbacks, such as low potency, lowered solubility and stability, as well as poor pharmacokinetics. Therefore, further studies are required to promote the phytochemicals’ bioavailability with novel formulations or by developing analogs with more potency, including nano-based drug delivery schemes [396], chitosan–pectin–core–shell nanoparticles loaded with phytochemicals [397], glycosylation agents [398], and encapsulated phytochemicals [399], to augment the polarity and reduce the poor pharmacokinetic profiles of natural phytochemicals, thereby enhancing their efficacy against cancer and driving their conversion from laboratory to bedside level. In vivo and in vitro studies on modulation of autophagy and apoptosis mediated by phytochemicals are required. Furthermore, an integrated systemic computational and pharmacological approach could be used to better comprehend the anticancer properties of phytochemicals.

Although production of plant-based monoclonal antibodies is advantageous in the treatment of cancer as stated above, there are significant caveats in their successful production. Protein G- or protein A-based affinity chromatography has been used to purify antibodies that have been expressed in plant systems [400]. However, to achieve purification, the plant tissues have to be homogenized first in order to break open the cell walls to release cell debris, contaminants, and noxious substances, following which these have to be removed by means of various purification steps [262]. However, there are challenges to protein purification because of problems of clogging in the chromatography columns caused by the cell wall debris that is left over in the process of homogenization of the plant biomass and removal of impurities [400]. Additionally, the application of protein A columns is circumscribed by its prohibitively high costs.

As regards plant viral VLPs and VNPs, although they are highly efficacious as anti-cancer vehicles, there exist several drawbacks that need to be addressed. In some instances, VNP/VLP capsid formation would likely be impaired due to the fusion of antigenic peptides with the respective viral capsid proteins. The necessity of efficacious infection and generation of VNPs in plants circumscribes the expression of epitopes, as the latter could result in the disruption of viral assembly and infection [10]. Additionally, procurement of reagents acceptable to the regulatory bodies poses several challenges. For manufacturing purposes, further research must evaluate optimization of codons between the chosen peptides and codon usage in plants to preclude the accumulation of unstable proteins [401,402]. Moreover, promoter usage, selection of untranslated regions, and thylakoid localization have to be factored in [403]. VNPs possess fundamental characteristics present in most classes of nanoparticles that can impact their viability in vivo [404,405]. Biological in vivo barriers, including interactions with antibodies, immune cells, and serum, could negatively impact the use of native or functionalized VNPs under clinical settings [406]. Also, in the human circulatory system, the surface of VNPs could be covered with proteins from the serum, resulting in protein corona that augment their ingestion by phagocytic cells [407,408]. The formation of protein corona can be an impediment in the development of VNPs for in vivo uses. Antibodies against the VNPs could alter interactions of VNPs with the immune cells, resulting in their elimination before they reach their target sites [409]. However, challenges to VNP delivery can usually be avoided or decreased by in situ vaccination. If the use of VNPs involves multiple VNP applications over many weeks, then it is likely that anti-VNP antibodies could be generated in the immune system that could restrict their half-life in the circulatory system and augment their elimination from the body [410,411]. The surface of the VNPs and VLPs could be coated with polyethylene glycol or other polymers [412] or albumin camouflaging [413] to circumvent this concern. Multiple treatments of viral nanocarriers can be precluded using implants, patches, and scaffolds that enable slow release of these particles.

## 20. Regulatory Aspects of Plant-Made Biopharmaceuticals

In general, it is well-established that drugs such as anticancer compounds are required to go through phase III clinical trials to obtain market permissions. The guidelines provided by the European Medicines Agency (EMA) and the Food and Drug Administration (FDA) specify that at least a single controlled Phase III trial having statistically notable results is necessary for allowing the license to market the respective plant-based biopharmaceutical product [414,415]. Excepting extraordinary circumstances, all these drugs must go through all clinical trial phases as per the EMA and FDA regulatory guidelines. Nevertheless, it has been noted that several pharmaceutical organizations deviate from established protocols and commence testing of the new drugs on human subjects prior to the defined timeline. The reason for these practices is to expedite the approval of the respective compounds due to pressure from investors [415]. Therefore, such anticancer drugs are presented for approval despite having insufficient information on their safety, efficacy, and quality. Although plant-derived biopharmaceuticals have been shown to have lesser toxicity when compared to traditional synthetic compounds, evidence has emerged regarding the undesirable side effects of unbridled and unregulated use of these compounds against cancer and other diseases. For example, *Fagonia cretica* has shown robust anticancer activity to breast cancer upon testing in the MDA-MB-231 cell line [416]. Moreover, *F. cretica* has been used in traditional medicine to treat several disorders, and some people have even used it as herbal tea to treat breast cancer. Nevertheless, there have been only a few reports on its anticancer activity.

Worldwide, the process of oncological drug development and sale is regulated by the recruitment of experts as well as advisory procedures mediated by the regulatory authorities [417]. There are many models of regulatory frameworks obtainable for prescribing these drugs, but harmony is required amongst regulatory bodies and enhancement in the regulatory process. The FDA has adopted guidelines for the regulation of the International Council for Harmonization regarding nonclinical assessment of drugs intended for cancer treatment. Regulatory authorities must be harmonious with other agencies to regulate plant-based anticancer compounds and should enhance the focus on integrating information obtained from traditional knowledge with scientific investigations regarding these drugs [418].

Furthermore, plants belonging to the same species when cultivated in different areas show variation in their content of medicinal compounds [419]. This calls for the necessity to focus on the generation of plants having enhanced qualities with uniform profiles of metabolites that, once investigated, are deemed safe or not conclusively. This can be achieved by the in vitro growth and continuing genetic and biotechnological studies on such anticancer plants [420,421].

## 21. Conclusions and Future Perspectives

Chemoprevention by means of dietary phytochemicals is a viable clinical approach in the management of carcinogenesis due to its simplicity and low cost. The multimodal clinical usage of phytochemicals in the form of multifunctional compounds is highly propitious due to the capability of these compounds to reverse or stop premalignant cells’ neoplastic transformation on a genomic level while preserving healthy cells and precluding the appearance of tumor phenotypes [422,423]. Phytochemicals, primarily as natural mixtures forming part of whole plant foods, provide effective antioxidant activities and function as principal chemopreventive agents during the initiation phase of neoplastic transformation [424]. The oncostatic roles of phytochemicals in the scavenging of free radicals, augmented endogenous biosynthesis of antioxidant enzymes, enhanced DNA repair, metabolic inactivation of carcinogens, inhibition of pro-oxidant enzymes, and detoxification have been well documented [425,426,427,428]. As phytochemicals are able to interfere with molecular mechanisms associated with the growth of the tumors and their metastatic spread, chemoprevention strategies must be designed to preclude the initiation of cancer while suppressing angiogenesis, cancer cell proliferation, and formation of malignant stem cells in addition to promoting apoptosis, regulation of epigenetic mechanisms, and immunity [422,429,430,431,432,433,434,435].

The mechanistic data on the benefits of phytochemicals at the preclinical level could be combined with clinical studies to provide clinical recommendations on the use of these substances in the chemoprevention of malignant cancer at the initial, secondary, and tertiary levels [436]. The pivotal aim of chemoprevention is the suppression of cancer incidence and progression. Evidently, the proclivity for plant-derived functional foods compared to single phytochemicals could provide the logical and efficacious approach to manage malignant diseases [424,437]. Nevertheless, there is an obvious paucity in terms of results substantiating these discoveries in clinical settings.

Navigating the challenges and opportunities presented by integrating phototherapy with plant-derived bioactive compounds opens the door to exciting future directions in cancer research and treatment. Further, elucidating the molecular mechanisms, optimizing treatment protocols, and conducting large-scale and rigorous clinical trials are imperative steps in realizing the full potential of this advanced approach. The dynamic synergy between light-based therapies and the rich pharmacopeia of plant-derived compounds holds promise for revolutionizing cancer treatment. As we venture into uncharted territories, integrating plant-derived compounds into phototherapy offers enhanced efficacy and the potential for minimizing adverse effects, paving the way for a new era in personalized and effective cancer therapeutics.

Plants function as propitious biofactories for antibody production on a large scale, owing to their low cost, increased production capacity, improved scalability, and facile growth procedures subject to containment conditions while dispensing with the need to screen for pathogens. Several plant-based MABs have been expressed on large scales in accordance with cGMP regulations and have proven to meet up to the US FDA standards of purity, potency, and identity, in addition to demonstrating appropriate assembly, efficacious neutralization in vitro, and potent efficacy in vivo in animal model systems [265]. The FDA has provided approval for over a dozen MABs meant for therapeutics of several malignancies, and many companies have been involved in the production of these molecules. Recent advancements in glycoengineering, posttranslational modifications, and genetic engineering have afforded further advantages beyond the conventional benefits of economic feasibility, augmented safety, and improved scalability, and it is estimated that plants can function as excellent systems for MAB production in the future.

Nanoparticles are increasingly used in the treatment of cancer with specific focus on therapeutic agent delivery. Nevertheless, administration of nanoparticles systemically has revealed that only about 1% of these particles can accumulate within the tumor microenvironment. Therefore, to make a significant impact, novel delivery systems are warranted. VNPs serve as building blocks, having capability for therapeutic agent delivery or for in situ vaccinations as well as in their combined forms. Multifunctional VNPs capable of loading, protecting, and regulating targeted cargo release, in addition to being intrinsically immunomodulatory, can enable immunogenic death of tumor cells, thereby modifying the tumor microenvironment. VNPs can be used for in situ vaccinations along with other multiple therapeutic strategies within a single platform. Such combination therapies could pave the way for efficacious inhibition of tumors.

Combination of VNPs and VLPs with radiotherapy and chemotherapy can potentiate anticancer responses by augmenting immunogenic cell death and increasing the pool of tumor-targeting T cells to promote their efficacy. With growing interest in this field, the count of immunotherapies based on plant viruses in pre-clinical trials is poised to expand and lead to clinical studies and regulatory approval of these plant virus-derived therapies to enhance cancer immunotherapy. A major number of scientific investigations have been conducted on the efficacy of plant viral nanoparticles in animal models or in vitro cell culture systems. Therefore, the next principal challenge would be the design and execution of clinical trials in humans. Additionally, mechanisms of delivery of VNPs into patients have to be investigated in further detail, and finally, regulatory policies have to be formulated to ensure the safe transition of VNP applicability from the bench level into the clinical phase [438].

Innovations in the manufacturing framework, financial backing, and maturation of regulatory structure regarding plant-based pharmaceuticals are vital to their commercialization in developing nations. As a majority of plant-derived biotechnology research and development has been sourced from academia, a transition from this phase to the corporate scenario will take tremendous effort at the very least [439]. Nevertheless, favorable features, including minimal production costs, lesser purification steps, along with increased levels of efficacy and safety, make plants an ideal platform for generation of monoclonal antibodies, viral nanoparticles, phytochemicals, and other related biopharmaceuticals. Ultimately, advanced medical strategies founded on personalized, pre-emptive, and predictive medicine are deemed the future of cancer management.

## Figures and Tables

**Figure 1 bioengineering-12-00007-f001:**
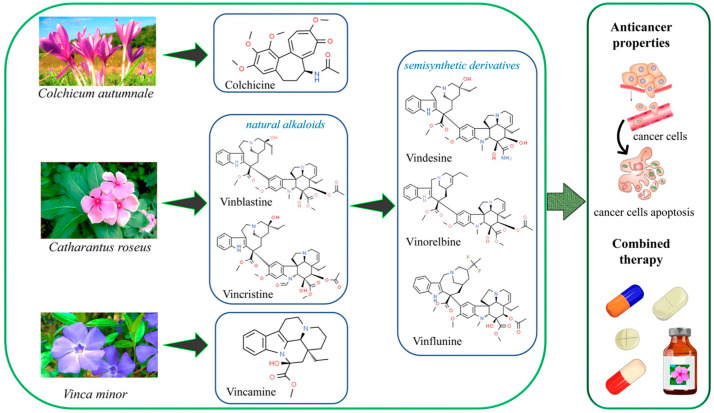
The chemical structures of some prominent natural alkaloids and their semisynthetic derivatives serve as effective agents in combating cancer. Reproduced from an open-access source Dhyani et al., 2022 [151].

**Figure 2 bioengineering-12-00007-f002:**
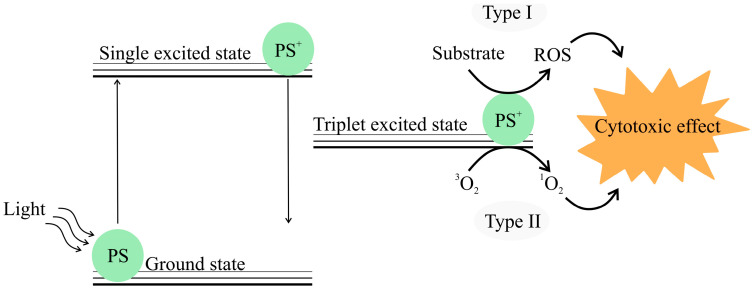
The phototherapy mechanism of action. In photodynamic therapy (PDT), photosensitizers (PS) absorb light, transitioning to an excited state. This leads to two pathways: PDT Type I, where the PS reacts with biomolecules to create reactive oxygen species (ROS), and PDT Type II, where the PS transfers energy directly to oxygen, producing ROS. ROS exhibits high oxidizing power, causing cytotoxic effects primarily near their site of generation due to their short lifespan. PS* refers to the photosensitizer’s excited state. Reproduced from an open access source Pivetta et al., 2021 [212].

**Figure 3 bioengineering-12-00007-f003:**
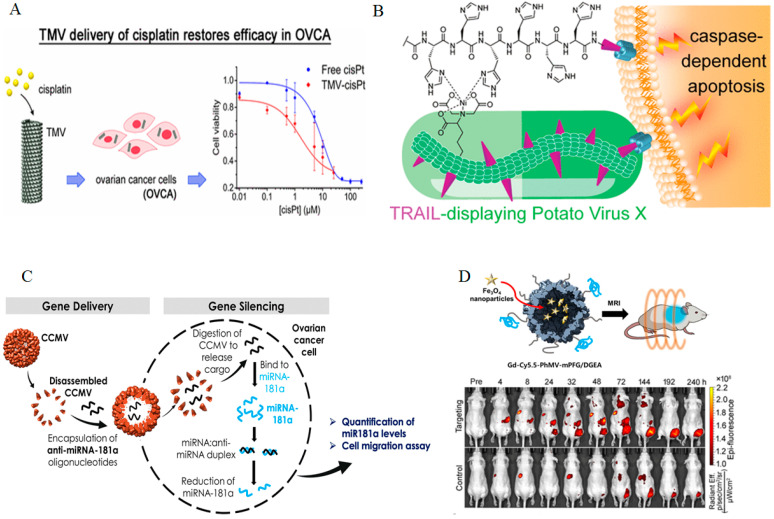
PVNPs as delivery therapeutic and imaging agents in cancer. (**A**) Tobacco mosaic virus (TMV) for the targeted delivery of cisplatin in Pt-resistant ovarian cancer cells [312] (Reprinted/Adapted with permission from [272] Copyright© 2018, American Chemical Society. (**B**) The preparative process for potato virus X (PVX)-HisTRAIL by coordinating the bond between a Ni-nitrilotriacetic (NTA) group on the virus; the His-tag at the N-terminus of HisTRAIL is shown with a purple triangle. Multivalent display of HisTRAIL on the elongated PVX particle permits proper binding on death receptors DR4/5 (the trimers with blue color) for activating the caspase-dependent apoptosis in cancerous cells [313] (Reprinted/Adapted with permission from [273] Copyright© 2019, American Chemical Society). (**C**) miR-181a is an important target for ovarian cancer therapy. qPCR data and cancer cell migration assays demonstrated higher knockdown efficacy when anti-miR-181a oligonucleotides were encapsulated and delivered using the VLPs resulting in reduced cancer cell invasiveness [314] [Adapted from open access source: 274 Citation needed]. (**D**) Schematic illustration of Gd-Cy5.5-PhMV-mPEG NPs for cancer imaging. In vivo NIR fluorescence images of PC-3 prostate tumors in athymic nude mice after the intravenous injection of Gd-Cy5.5-PhMV-DGEA [315] [Adapted from open access source 275: Citation needed].

**Figure 4 bioengineering-12-00007-f004:**
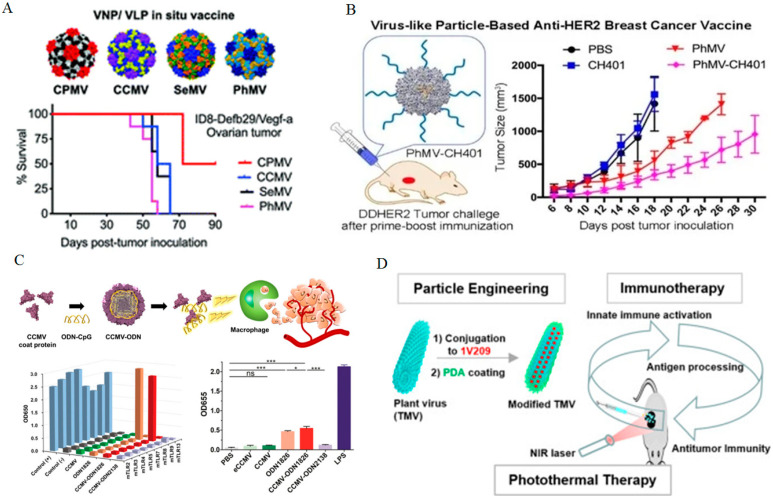
PVNPs in cancer immune and combinational therapy (**A**) Intratumoral administration of plant-derived Cowpea mosaic virus (CPMV) nanoparticles as an in situ vaccine overcomes the local immunosuppression and stimulates a potent anti-tumor response in several mouse cancer models and canine patients [349] (Adapted from open access source: 309, Citation needed). (**B**) The PhMV-based anti-HER2 vaccine PhMV-CH401, demonstrated efficacy as an anti-HER2 cancer vaccine. Our studies highlight that VLPs derived from PhMV are a promising platform to develop cancer vaccines [350] (Adapted from open access source: 310, Citation needed). (**C**) Schematic diagram of preparing CCMV VLPs containing ODN 1826 (CCMV-ODN1826) for cancer therapy [315] (Adapted from open access source: 275, Citation needed). (**D**) Photothermal immunotherapy of melanoma using TLR-7 agonist laden TMV with polydopamine coat [325]. (Adapted from open access source: 285, Citation needed). Statistical significance was measured by one-way ANOVA with Tukey’s test: * *p* < 0.05, ** *p* < 0.01, *** *p* < 0.001. ns refers to not significant.

**Table 1 bioengineering-12-00007-t001:** Examples of phytochemicals and their anticancer effects.

Phytochemicals and Plant Source	Type of Cancer	Activity/Effects
Artemisinin (sesquiterpene lactone) *Artemisia annua*	Human breast cancer cells	pH-sensitive nanoparticles (D/D NPs) containing dihydroartemisinin (DHA) and docetaxel (DTX) demonstrate anticancer activity in vitro in breast cancer cells by augmenting ROS, decreasing mitochondrial membrane potential, enhancing p53 expression, and eliciting release of cytochrome C into the cytoplasm that in turn activates caspase-3 [33].
Triterpene 3,7-dihydroxy-25-methoxycucurbita-5,23-diene-19-al of the cucurbitane-type (DMC) *Momordica charantia* L.	Human breast cancer cells	Inhibits mTOR-p70S6K signaling through activation of AMPK and downregulation of Akt, resulting in cytoprotective autophagy [34].
Furanodiene (sesquiterpene) *Rhizoma curcumae*	Lung cancer cell lines	Blocks cell proliferation and inhibits progression of the cell cycle in the G1 phase through downregulation of CDK6 and cyclin D1 protein levels and upregulation of the levels of p27 and p21 in 95-D cells [35]. Downregulates the levels of Bcl-2, survivin, pro-caspase-7, and full PARP and upregulates the level of cleaved PARP. It augments the light chain 3-II (LC3-II) protein levels, implicating the involvement of autophagy.
Berberine (benzylisoquinoline alkaloid) *Berberis vulgaris*	Human gastric cancer cells in vivo and in vitro	Elicits cytostatic autophagy through inhibition of Akt and MAPK/mTOR/p70S6K pathways [36]. Notably downregulates HIF-1 and vascular endothelial growth factor (VEGF) expression, which reverses resistance to radiotherapy [37].
Caffeine Coffee beans	Human lung adenocarcinoma spheroid models	Decreases the expression of Nrf2 and Claudin-2, leading to impairment of mitochondrial respiration and generation of ROS [38]. Exaggerates cisplatin and doxorubicin toxicity in these spheroids [38].
Deguelin Peas and beans of the Leguminosae family	Lung cancer cells	Induces apoptosis by augmenting release of cytochrome C and protein levels of the apoptosis induction factors. Elicits PUMA expression [39] and inhibits the P13K/AKT pathway, triggering the binding of Foxo3a with the PUMA promoter to induce its transcription. In turn, PUMA stimulates both Bax and the intrinsic mitochondrial cellular death pathway. Augments doxorubicin chemotherapeutic sensitivity in vivo and in vitro.
Piperlongumin *Piper longum* L.		Regulates STAT3 (signal transducer and activator of transcription 3), nuclear factor kappa B, phosphatidylino-sitol 3-kinase/protein kinase B, cyclooxygenase-2, cyclin D1, and the glutathione pathway, which are involved in cancer initiation, cellular proliferation, and tumor progression [40,41,42,43,44,45,46,47]. Induces antioxidant and immunity-promoting effects.
Vinblastine *Catharantus roseus*	Ovarian cancer, breast cancer, osteosarcoma, lung carcinoma, lymphoma, leukemia, and gastric cancer A549, CCF-STTG1, HGC-27, Hela, and MCF-7 cancer cell lines and murine cancer models	Microtubule targeting agent that disrupts microtubule polymerization; when loaded on to graphene quantum dots (GQD), enhanced its cytotoxicty in cancer cells while at the same time exhibiting lowered toxicity towards normal cells [48,49,50].
Vincristine *Catharantus roseus*	Acute lymphoblastic leukemia (ALL) in vitro and in vivo models	Daratumumab–polymersome–vincristine (DP-VCR) shows high selectivity for CD38+ ALL cells in vitro [51]. In vivo murine models treated with DP-VCR showed significant decrease in leukemia burden in the liver, spleen, blood, and bone marrow and improved survival along with fewer side effects. This proved DP-VCR to be a potent and safe nanotherapy for CD38+ ALL.
Piperine *Piper nigrum*	Colon cancer cell lines	Suppresses glucuronidation of many chemopreventive substances, resulting in enhancement of their bioavailability; inhibits the cell cycle; and promotes apoptosis [52].
Aged black garlic extract *Allium sativum* S-Allyl-Mercapto-Cysteine (SAMC) Black garlic extract	MDA-MB-361 and MCF-7 cell line ER+ breast cancer cells U937 human leukemia cells HT29 colon cancer cells SW620 human colon cancer cell line Mouse macrophage line (TIB-71), MCF-7 breast cancer, prostate cancer (PC-3), and Hep-G2 cells HL-60 leukemia cells Human gastric cancer cells SGC-7901 and in murine models	Induces apoptosis of these cells through blockage of BCL-2 and MCL-1 anti-apoptotic protein expression while eliciting the expression of BAK and BIM pro-apoptotic proteins [53]. The decrease in MCL-1 expression is mediated by the activation of JNK caused by an enhancement of ROS production in cancer cells [53]. Stimulates caspase-based apoptosis initiated by both extrinsic and intrinsic pathways [54]. Blocks proliferation and stimulates apoptosis, possibly by modulating the PI3K/Akt signaling pathway, promoting PTEN expression, and decreasing Akt and p-Akt expression [55]. Induces apoptosis via the p38 and JNK pathways, which in turn activate the p53 and Bax [56]. Inhibits cell cycle and cell proliferation, finally leading to apoptosis [57,58]. Dose-determined cytotoxic effects [59]. Exhibits immunomodulatory and anticancer effects wherein ABGE augmented GSH-Px and SOD activity and led to apoptosis and inhibition of cancer cell growth [60].
Sulforaphane Cruciferous vegetables, including broccoli and brussels sprouts	Breast cancer stem cells and triple negative breast cancer cells	Exhibits anti-inflammatory and antioxidant potentials and represses the growth of cancer and associated cell-proliferative capabilities. Particularly, Notch and wnt/β-catenin BCSC-associated pathways are abrogated [61].
Isothiocyanates Cruciferous vegetables	Colon, liver, breast, prostate, bladder, pancreatic, lung, endometrial, and glioblastoma cancer	Induces anticancer activities [62,63,64,65,66,67,68,69,70,71]
Curcumin (polyphenol) *Curcuma longa*	Myeloid leukemia cell line HT-29 and AGS human cancer cell lines Glioma cell lines Breast cancer stem cells (BCSCs) Head and neck cancer cells	Elicits apoptosis and autophagy by negative regulation of the Bcl-2 protein [72]. Induces ER stress and malfunction of mitochondria to trigger apoptosis [73]. Increases expression of ING4 and p21, following which it upregulates BAX and downregulates the NF-B and Bcl-2 signaling pathways resulting in apoptosis [74,75]. Terminates the recognized wnt/β-catenin pathway, thus precluding β-catenin nuclear translocation and Slug transcription factor activation. This leads to restoration of the expression of E-cadherin and blockage of BCSC and EMT migration [76]. Inhibits the PI3K/Akt/mTOR pathway [77], nuclear factor-kappa B (NF-κB), and p38 mitogen-activated protein kinase (MAPK) pathways [78].
Epigallocatechin-3-gallate (EGCG), (polyphenol) *Camellia sinensis* (green tea)	Breast cancer cells ER+ breast cancer cells	Blocks proliferation of tumor cells, triggers apoptosis, precludes angiogenesis and cytokine synthesis [79], blocks the proliferative and angiogenic capability of breast cancer cells, wherein it inhibits the expression of hypoxiainducible factor 1 subunit alpha (HIF-1α), activates NF-kB, and expresses vascular endothelial growth factor (VEGF) in mouse models [80,81]. Downregulates matrix metalloproteinase-9 (MMP9) induced by EGF, resulting in cessation of metastasis and cellular invasion [82]; interferes with the PI3K/Akt pathway, impeding cancer cell survival and proliferation [83,84].
Gallic acid Onions, red fruits, and tea	Lung cancer cells and murine models of lung cancer	Gallic acid and cisplatin block colony formation and formation of tumor spheroids [85], elicit apoptosis, and inhibit the P13K/Akt pathway, which upregulates the tumor suppressor protein, p53, that, in turn, controls proteins related to the cell cycle, such as E1, Cyclin D1, p21, and p27 as well as intrinsic apoptotic proteins, including cleaved caspase-3, Bax, and Bcl-2. Blocks progression of lung cancer by arresting cell cycle and inducing apoptosis, thereby making it a promising therapeutic candidate to confront non-small cell lung cancer; functions as an adjuvant to promote the cytotoxicity of cisplatin towards lung cancer cells [85]
Honokiol (biphenolic neolignane) *Magnolia officinalis*	Human glioblastoma	Blocks glioblastoma cell proliferation by inciting slight arrest of the G0/G1 phase cell cycle and causing apoptosis through both caspase-dependent and caspase-independent pathways [86]; the apoptotic effect involves blockage of STAT3 signaling and ERK1/2 in addition to the activation of p38 MAPK signaling pathway.
Oleocanthal Virgin olive oils	Prostate and breast cancer cells	Suppresses proliferation, invasion, and migration of cancer cells by inhibiting c-Met phosphorylation, blocks progression of the cell cycle as well as cell proliferation, elicits oxidative stress, and induces apoptosis while stimulating the immune system, thereby precluding carcinogenesis [87].
Cinnamaldehyde from *Cinnaomomum* species and chlorogenic acid from green coffee	Breast cancer cell lines and experimental models of breast cancer	Blocks initiation of tumor formation by detoxifying carcinogens, preventing the formation of DNA adducts, scouring electrophilic species, preventing peroxidation of lipids, and protecting against mutagenesis; tumor suppressing by inhibiting the growth of preneoplastic tissues, vascularization, capability of stemness, tumor metastasis and invasion, promoting autophagy and apoptosis, repressing tumor cell invasion and migration, disrupting the energy metabolism of cancerous tissues, and blocking estrogen receptors [88,89,90,91].
Resveratrol Grapes, blueberries and cranberries	Ovarian cancer cell lines Breast cancer stem cells	Initiates autophagy, wherein it lowers the amount of mTOR and phosphorylated Akt [92]. Induces the downregulation of the Wnt/β-Catenin signaling pathway and causes autophagy; has antitumor, antioxidant, and anti-inflammatory properties [93].
Anacardic acid (2-hydroxy-6-pentadecylbenzoic acid) Cashew nut shells	A549 human lung cancer cells	Induces ER stress, which promotes CHOP expression as well as cleavage of caspase-12 in addition to the disruption of Ca^2+^ homeostasis, resulting in apoptosis [94].
Ampelopsin *Ampelopsis grossedentata*	MCF-7 and MDA-MB-231 breast cancer cells	Induces intracellular ROS production and apoptosis associated with malfunction of mitochondria in breast cancer cells including loss of mitochondrial membrane potential, build-up of high levels of ROS and augmented expression of Bcl-2/Bax expression [95].
Apigenin Bell pepper, garlic, cabbage and celery	A549 lung cancer cells	Inhibits cell growth and promotes apoptosis likely though enhancement of ROS generation while having no effect on normal cells, following which caspases 3 and 9 are induced, leading to the death of A549 cells through apoptosis [96].
Artocarpin *Artocarpus* species	Non-small cell lung carcinoma (NSCLC, A549) cell lines	Phosphorylates and activates cellular protein kinases AktS473, p38, and Erk1/2, followed by apoptosis mediated by the elicitation of ROS [97]; activates p53-dependent apoptotic proteins Apaf-1, caspase-3, cytochrome c, and PUMA; elicits apoptosis mediated by the augmentation of both independent AktS473/NF-κB/c-Myc/Noxa and ERK/p38/p53-dependent cascades by ROS.
Butein *Butea monosperma*	Human ovarian cancer cells and mouse xenografts	Inhibits interaction between IL-6/IL-6Rα and regulates the IL-6/STAT3/FoxO3a pathway; reduces cell proliferation, invasion, and migration in addition to enhancement of apoptosis and cell cycle arrest [98].
Chrysin Blue passionflower, propolis and honey	In vivo tumor models and cancer cell lines	Inhibits tumor growth by inducing apoptosis, altering cell cycle, inhibiting invasion, angiogenesis, and metastasis while being non-toxic to normal, healthy cells; augments the ratio of Bax/Bcl2, induces caspases 3 and 9, and stimulates lung cancer cell apoptosis [99].
Delphinidin Bilberry (*Vaccinium myrtillus*), jamun (*Syzygium cumini*), and blackcurrant (*Ribes nigrum*)	HCT116 human colon cancer cells	Elicits apoptosis through the generation of ROS, activates cytochrome C, caspase 3, 8, and 9, and pro-apoptotic Bax and inhibits the expression of anti-apoptotic proteins, including ERK1/2, p38, and STAT-3 [100].
Genistein Soybeans	Ovarian cancer cells, prostate and breast cancer	Induces cell death in cancer cells via the caspase-independent pathway by inhibiting glucose uptake and leading to autophagy and apoptosis [101,102,103,104,105,106,107]; induces apoptotic effects by modulating the Fas-FasL pathway, TRAIL-DR pathway, TNF-α-TNFR1 pathway, Bcl2-Bax pathway, and targeting the PI3K-Akt-mTOR pathway and the JAK-STAT3 signal pathway. It exerts notable antiproliferative activities against ER+ human breast cancer cells by inducing G2-M arrest, p21 expression, followed by apoptosis.
Kuwanon Mulberry root	NCI-H292 and A549 lung cancer cells	Decreases cell migration and proliferation, while augmenting apoptosis via the mitochondrial pathway, paraptosis by incrementing cytoplasmic vacuolation and by inducing ER stress [108,109]
Quercetin Many vegetables, onion, apples, green tea, berries and red wine	Gastric cancer cells Triple-negative and ER+ breast cancer cell lines Colorectal adenocarcinoma and hepatocellular cell lines Breast cancer cells	Augments the accretion of hypoxia-induced factor 1 (HIF-1), in turn inhibiting mTOR signaling and stimulating the biosynthesis of BNIP3/BNIP3L. This process disrupts the Beclin 1/Bcl-2 (Bcl-xL) complex, leading to the activation of autophagy [110]. Initiates apoptotic cell death [111]. Quercetin–zinc(II) complex induces apoptosis [112,113]. Downregulates ALDH1A1 activity; suppresses Mucin 1 (MUC1) expression by inhibiting cell proliferation and cancer metastasis; downregulates the expression of epithelial cell adhesion molecule (EpCAM) implicated to be actively involved in inducing cancer stemness, cellular proliferation, angiogenesis, metabolism, drug resistance, and epithelial to mesenchymal transition (EMT); shuts down the stemness of breast tumor progenitor cells [114].
Silibinin *Silybum marianum* (milk thistle) and *Cynara scolymus* (artichoke)	Breast cancer cells	Interacts with Erα and influences RAS/ERK and P13K/AKT/mTOR pathways of signal transduction, thereby inducing autophagy. Its interaction with Erβ enhances apoptosis. Silibinin blocks metastasis through EMT suppression by inhibiting the expression of TGF-β2. The anti-metastatic effects of silibinin is also associated with the Jak2/STAT3 pathway [115].
Juglone *Carya catharsis*	Human endometrial cancer cells	Upregulates the expression of p21 mRNA and protein, concomitant with diminished levels of cyclin A, CHK1, cdc25A, and CDK2. Furthermore, it leads to the downregulation of Bcl-xL and Bcl-2 and upregulation of cytochrome C, Bax, and Bad, suggesting its association with the mitochondrial pathway during apoptosis [116].
2-Methoxy6acetyl7methyljuglone (MAM) *Polygonum cuspidatum*	A549 lung cancer cells	Results in necroptosis and production of nitric oxide through activation of JNK; this augments peroxidation of lipids, leading to the generation of peroxynitrite (ONOO–), which triggers apoptosis [117].
Dioscin *Polygonatum sibiricum*	Breast cancer	Reduces breast cancer stemness by arresting the cell cycle through regulation of AKT/mTOR and p38 MAPK signaling pathways [118]. Dioscin elicits the expression of p53 and p21 and blocks the expression of many cyclin-dependent kinases and cyclins.
Ginsenosides *Panax ginseng*	Cancer cells	Controls the p53 pathway, neutralizes ROS, modulates miRNAs through decrease in Smad2 expression, regulates Bcl-2 expression through NF-kB pathway normalization, blocks inflammatory pathways through reduction in cytokine production, incites cell cycle arrest by restriction of CDC2 and cyclin E1, and induces apoptosis of cancerous cells [119].
Garcinol *Garcinia indica*	SKBR3A, MDAMB231, and MCF7 breast cancer cell lines	Downregulates the expression of anti-apoptotic proteins like Bax and Bcl-XL; elicits cell cycle arrest followed by apoptosis in breast cancer cells overexpressing Her-2; causes loss of fragmentation of mitochondria and mitochondrial transmembrane potential, leading to apoptosis in MCF-7 cells [120].
Propolis Honey bees from substances collected from parts of plants, buds, and exudates	MCF-7 human breast cancer cells	Causes ER stress whereupon CCAAT/enhancer binding homologous protein (CHOP) in turn elicits apoptosis in response to the ER stress [121].
Thymoquinone *Nigella sativa*	Head and neck squamous cell cancer cells Oral cancer cells and breast cancer cells	Elicits cell death through autophagy dependent on LC3-II activation and apoptosis dependent on caspase activation [122]; causes strong cytotoxicity; and incites apoptotic cell death, as shown by increased caspase-9 activation and Bax expression. Causes cell death by means of anti-neoplastic effects that can elicit autophagy and apoptosis; blocks bone metastasis associated with breast cancer cells by mediating disruption of NF-kB and CXCR4 signaling axis [123].
6-Shogaol *Zingiber officinale* Rosc	SMMC-7721 cells (human hepatocellular carcinoma cell line)	Induces ER stress; PERK/eIF2α dephosphorylation and induction of the expression of the downstream CHOP generate a caspase cascade effect that results in apoptosis [124].
γ–Tocotrienol (Vitamin E) Annatto seeds, palm oil and rice bran oil	LNCaP and PC-3 human prostate cancer cells	Elicits autophagy, apoptosis and necrosis accompanied by enhancement of intracellular dihydrosphingosine and dihydroceramide levels. indicating modulation of the sphingolipid biosynthetic pathway [125].
ω-Hydroxyundec-9-enoic acid (ω-HUA) *Oryza officinalis*	Human non-small cell lung cancer (NSCLC)	Induces ROS whereupon biosynthesis of CHOP and phosphorylated p-eIF2α were suppressed by ROS along with NAC, revealing that ROS is vital for x-HUA-stimulated ER stress and caspase-enabled apoptosis [126]
Carotenoids such as lutein, lycopene, zeaxanthin, α-carotene, β-Carotene and astaxanthin [127] Vegetables, fruits, milk, meats, eggs, some crustacean seafoods and fish [128,129] Saffron extract from *Crocus sativus*	Cancer cells Liver cancer	Suppress the biosynthesis of pro-inflammatory cytokine molecules as well as enzymes, including COX-2 and NO in LPS-elicited cells; further, carotenoid extract showed anti-inflammatory potential by inhibiting JNK phosphorylation and NF-κB activation [130]. Decreases cell proliferation, oxidative stress, and inflammation; elicits apoptosis in addition to the downregulation of inflammatory markers like NF-κB-p65, iNOS, and COX-2 in vivo [131].
Emodin *Rheum officinale* and *Polygonum cuspidatum*	Triple-negative breast cancer (TNBC) murine models MDA-MB-435S cells in vitro and in mouse models Colorectal cancer	Targets transcriptional regulators SerRS and NCOR2 to inhibit the transcription of anti-vascular endothelial growth factor A (VEGFA) as well as tumor angiogenesis in murine models [132]. Emodin liposomes and daunorubicin liposomes modified with arginine_8_-glycine-aspartic acid (R_8_GD) were strongly cytotoxic and efficiently suppressed the generation of VM (vasculogenic mimicry) channels and tumor cell metastasis occurring in invasive breast cancer [133]. Additionally, they induced the downregulation of some metastasis-associated proteins such as HIF-1α, TGF-β1, VE-cad and MMP-2. Inhibits cell proliferation and elicits apoptosis [134]; reduces GSH content and expression of GPX4 and xCT while augmenting the generation of ROS, lipid peroxidation, and MDA; inactivates the NF-κb pathway in these cells and in murine models wherein it inhibited tumor growth and elicited in vivo ferroptosis through the inactivation of the NF-κb pathway.

**Table 2 bioengineering-12-00007-t002:** A few examples of the preparation and extraction methods of important phytochemicals.

Phytochemical	Preparation and Extraction Methods
Curcumin	Traditional methods using extraction with ethanol, distillation with steam, hot and cold percolation, utilizing alkaline solution [152] and hydrotrope [153]. Advanced methodologies such as extraction with supercritical fluid bereft of organic solvents, microwave and ultrasonic extraction, Soxhlet extraction, enzyme-mediated extraction [154,155]; chromatography for separation of curcuminoids from the co-extracted oleoresins and volatile oils, bisdemethoxycurcumin, and demethoxycurcumin [156,157].
Resveratrol	Extraction with organic solvents [158,159], enzyme-mediated ultrasonic extraction [160], maceration [161], thermal heating and subsequent enzyme treatment of grape peel extracts with pectinases and glucanase [162]; solid-phase extraction HPLC methodology coupled with a nanofibrous sorbent [163]; quick magnetic solid phase extraction using mesoporous nanoparticles grafted with alendronate sodium to efficiently identify trans-resveratrol [164].
EGCG (Epigallocatechin Gallate)	Traditional solvent extraction, ultrasound-enabled extraction, microwave-assisted extraction, supercritical CO_2_, Soxhlet extraction, processing under high pressure, subcritical water extraction [165,166,167]; green extracting compound like cyclodextrin augmented the yield of EGCG [168]; polymeric electrode PAN/PPY laden with TiO_2_ and rGO nanoparticles improved the efficacy of extraction of EGCG to high purity [169].
Allicin	Supercritical CO2 extraction [170], pressurized liquid extraction [171], supercritical fluid extraction [172], ultrasonic assisted extraction [173]; HPLC-MTT assay [174], salting-out extraction [175]; water extraction followed by ultrasound-enabled binding with isolates from whey protein improved the solubility, stability, and emulsifying properties [176].
Emodin	Maceration, reflux extraction, microwave-enabled extraction, ultrasonication extraction, ultrasonic nebulization extraction, stirring extraction, preparative liquid chromatography, and supercritical CO_2_ extraction [177,178,179,180].
Genistein	Treatment with enzymes and/or acid followed by extraction with solvent [181], ultrasonication [182], extraction with supercritical fluid with and without enzyme hydrolysis [183,184]; chemical synthesis utilizing microwave ovens [185], germinating soybean seeds, and transgenic rice with high content of genistein [185].
Parthenolide	Feverfew extraction using petroleum ether and chloroform [186], gradient HPLC [187], and supercritical CO_2_ extraction; bottle stirring methods using acetonitrile with 10% water (*v*/*v*) gave the highest yield [188,189,190].
Luteolin	Maceration, followed by heat reflux Soxhlet extraction; reflux with methanol proved to be superior to other extraction techniques [191]; other techniques include hydrodistillation [192] ultrasonic-assisted extraction [193], microwave-assisted method [194], and enzyme-assisted extraction [195].
Quercitin	Simple ethyl acetate cold extraction [196], supercritical CO_2_ extraction [197], ultrasound-assisted extraction [198], subcritical water extraction [199], microwave extraction [200], and ionic liquid-based extraction with pressurized liquid combined with HPLC [201].

**Table 3 bioengineering-12-00007-t003:** Clinical trials of phytochemicals against cancer.

Study	Drugs Involved	Conditions/Effects	Status	Identifier	References
Therapeutic effect of luteolin natural extract versus its nanoparticles on tongue squamous cell carcinoma cell line	Luteolin Nano-luteolin	Tongue neoplasms Carcinoma	Unknown	NCT03288298	[202]
Artemisinin derivative SM934 inhibits expression of cathepsin K after forming a complex with testosterone	SM934 (a novel water-soluble artemisinin analog)	Inhibits proliferation and metastasis in breast cancer	Phase II	NA	[203,204]
Study of Liposomal Curcumin in combination with RT and TMZ in patients with newly diagnosed high-grade gliomas	Curcumin combined with radiotherapy (RT) and Temozolomide (TMZ)	Glioblastoma	Phase I/Phase II	NCT05768919	[205]
Curcumin Bioavailability in Glioblastoma Patients	Curcumin	Glioblastoma	Unknown	NCT01712542	[206]
Phase I Assay-guided Trial of Anti-inflammatory Phytochemicals in Patients With Advanced Cancer	Grape seed extract and vitamin D	Solid cancers (gastrointestinal, lung, breast, prostate, lymphoma, or cancer of the lymph nodes)	Phase 1 Completed	NCT01820299	[207]
Dietary Intervention With Phytochemicals and Polyunsaturated Fatty Acids in Prostate Cancer Patients	Tomato or a multi-diet consisting of grape juice, pomegranate juice, tomato, green tea, black tea, soy, selenium, and PUFAs	Prostate cancer	Phase 1 and Phase 2 Completed	NCT00433797	[208]
Clinical Trial of Lung Cancer Chemoprevention With Sulforaphane in Former Smokers	Sulforaphane	Lung cancer	Phase 2 Completed	NCT03232138	[209]
Black Raspberry Confection in Preventing Oral Cancer in Healthy Volunteers	Black raspberry confection	Oral cancer	Phase 1, Active, not recruiting	NCT01961869	[209]
Docetaxel With a Phytochemical in Treating Patients With Hormone Independent Metastatic Prostate Cancer (PROTAXY)	Phytochemical dietary supplement with docetaxel	Prostate cancer	Phase 2, Completed	NCT01012141	[209]
Tangerine or Red Tomato Juice in Treating Patients With Prostate Cancer Undergoing Surgery	Tangerine tomato juice or red tomato juice rich in lycopene	Prostate cancer	Not applicable	NCT02144649	[209]

**Table 4 bioengineering-12-00007-t004:** Recent reports of anticancer MAB expression in plants.

Type of Cancer	Plant System Used	Effects	Reference
Colorectal cancer	Transgenic tobacco expressing large single chain (LSC) antibody CO17-1A (LSC CO) and LSC CO tagged with the endoplasmic reticulum (ER) retention signal KDEL (LSC COK)	In vitro binding activity towards human colon cancer cell lines	[268]
Breast cancer	Transgenic tobacco expressing anti-HER2 VHH-FcK MAB	Bound to cancer cells in vitro and inhibited cell migration	[269]
Colorectal cancer and breast cancer	Transgenic tobacco expressing both MABs LSC CO17-1AK and anti-HER2 VHH-FcK in the same plant	Demonstrated binding to human SW620 and SKBR-3 cancer cells and inhibition of cell migration in vitro	[270]
Mouse colorectal cancer	Transient expression of recombinant bispecific monoclonal antibody for dual inhibition of programmed cell death protein 1/programmed cell death ligand 1 and cytotoxic T-lymphocyte-associated protein 4 axes in *Nicotiana benthamiana*	Significant inhibition of tumor growth in vivo and reduction in tumor weight and volume	[267]
Murine colon cancer	Transient expression of anti-CTLA-4 2C8 MAB in *N. benthamiana* by agroinfiltration	Recognition and binding to both human and murine CTLA-4 in vitro as well as inhibition of in vivo tumor growth	[267]
Mouse colorectal tumor	Atezolizumab anti-PD-L1 antibody transiently produced in *N. benthamiana*	Mouse tumor growth inhibition and in vitro binding to PD-L1	[271]
Mouse MC38 colon cancer	Recombinant anti-PD-1 Nivolumab was produced in *Nicotiana benthamiana* by transgenic technology	Reduced in vivo mouse tumor growth	[272]
Gastric and colorectal cancer	Transient expression of Durvalumab variants in *Nicotiana benthamiana*	Recognition and binding to recombinant PD-L1 and to PD-L1 expressed in gastrointestinal cancer cells, precluding its interaction with PD-1 on T cells, thereby augmenting T-cell immunity	[273]
Breast cancer	Trastuzumab transgenically expressed in glycoengineered rice	Inhibition of BT-474 cancer cell line proliferation, increased ADCC efficacy against Jurkat cells, and efficacious tumor uptake with lower liver uptake compared to TMab in a xenograft assay using the BT-474 murine model.	[274]
Hodgkin lymphoma, melanoma, lung colorectal, and breast and cancer	Transient expression of pembrolizumab and nivolumab in *Nicotiana benthamiana*	PD-1/PD-L1 inhibitory activity in vitro; both immune checkpoint inhibitors (ICIs) inhibit the PD-1/PD-L1 immune checkpoint leading to CTL activation and the elicitation of apoptosis in tumorigenic cells via T-cell-mediated cytotoxicity	[275]
CD27-expressing lymphoma and leukemia, recurrent glioblastoma, advanced solid tumors	Transient generation of Varlilumab (anti-human CD27) in *N. benthamiana*	Co-expression with chimeric beta 1,4-GALT (beta 1,4-galactosyltransferase) successfully achieved biantennary b1,4-galactosylated Varlilumab	[276]

**Table 5 bioengineering-12-00007-t005:** Various therapeutic mechanisms of plant virus nanoparticles (PVNPs) in cancer.

Therapeutic Mechanism	Description	Example	Advantages	Challenges	Ref
Targeted Drug Delivery	PVNPs are engineered to deliver drugs directly to tumor cells, minimizing side effects and enhancing treatment efficacy.	Chemotherapy drugs encapsulated in PVNP	Reduces systemic toxicity—increases drug concentration at tumor site	Requires precise targeting	[292]
Gene Therapy	PVNPs deliver genetic material to correct or modify defective genes within cells.	siRNA delivered via PVNPs	Potential to cure genetic disorders—can provide long-term effects	Delivery efficiency—risk of off-target effects	[293]
Delivery of cancer antigens (vaccines)	PVNPs can stimulate the immune system to attack cancer cells	PVNPs loaded with cancer antigens	Harnesses body’s natural defenses—can provide long-lasting protection	Risk of autoimmune reactions—requires careful modulation of immune response	[294]
In situ Vaccination	PVNPs are used as vaccines directly at the tumor site, inducing a localized immune response against cancer cells.	Monotherapy with PVNPs or combined with tumor-associated antigens	Induces strong local immune response—minimizes systemic side effects	Requires precise administration—potential for local inflammation	[295,296]
Delivery of immunoadjuvants	PVNPs modulate the immune system to enhance its response to cancer.	PVNPs or loading with Toll-like receptor (TLR) agonists	Enhances efficacy of existing treatments—can overcome immune evasion by tumors	Risk of over-stimulation of the immune system—balancing immune activation and suppression	[297]
Combination Therapy	PVNPs are used in combination with other treatments (e.g., chemotherapy, radiation) to enhance overall therapeutic efficacy.	PVNPs combined with chemotherapy drugs, Immune Checkpoint Inhibitors	Synergistic effects—can target multiple pathways	Complexity of treatment regimen—potential for increased side effects	[298,299]

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
