# Peer review of "Plant-Derived Anti-Cancer Therapeutics and Biopharmaceuticals"

_bioengineering, 2024, doi:10.3390/bioengineering12010007_

Round 1
Reviewer 1 Report
Comments and Suggestions for Authors
The manuscript bioengineering-3340067 “Plant-derived anti-cancer therapeutics and biopharmaceuticals” by Ghyda Murad Hashim et al. discusses the anticancer activities of key phytochemicals and plant-derived anticancer vaccines, nanoparticles, monoclonal antibodies, immunotherapies, etc.
The topic of research is. The authors use the relevant references. Nevertheless, the article requires major revision before accepting for publication.
Comments and questions:
1) References to World Health Organisation statements on the global health challenge of the cancer threat should be added to the Introduction.
2) Sections 5-12 would be well summarized into an overall scheme that represents the different types of bioactive compounds and the types of cancer against which they are active.
3) The application of nanoparticles in cancer treatment should be described in more detail. The active targeting of nanoparticles and the passive targeting due to the EPR effect are important benefits of nanoformulations and they should be added in Section 14.
Author Response
Reviewer #1
We thank the reviewer for taking the time to review our manuscript and for the valuable comments. We present herewith our point-to-point rebuttal in response to the comments.
Comments and questions:
- References to World Health Organisation statements on the global health challenge of the cancer threat should be added to the Introduction.
Author reply: We thank you kindly for this comment. The WHO reference has been included in the Introduction.
- Sections 5-12 would be well summarized into an overall scheme that represents the different types of bioactive compounds and the types of cancer against which they are active.
Author reply: A table has been made enlisting the various phytochemicals, the types of cancer and their anticancer effects.
- The application of nanoparticles in cancer treatment should be described in more detail. The active targeting of nanoparticles and the passive targeting due to the EPR effect are important benefits of nanoformulations and they should be added in Section 14.
Author reply: Thanks for your comment. The required addition has been made to section 7 in the revised version of the manuscript.

Reviewer 2 Report
Comments and Suggestions for Authors
The manuscript “Plant-Derived Anti-Cancer Therapeutics and Biopharmaceuticals’ by Ghyda Hashim et al. is devoted to discussion of the anticancer activities of principal phytochemicals with focus on signaling circuits towards targeted cancer prophylaxis and therapy. The manuscript is a full-fledged review of modern literature, covering all the necessary aspects of the claimed research, and can be published in the journal Bioengineering after correcting a number of comments.
1. Lines 123-124: "problems of the mouth"; in my opinion, an incorrect phrase, perhaps it is better to use "problems with the oral cavity" or "problems with the oral mucosa"
2. Lines 144-146: "The delay in cancer diagnosis combined with non-responsive therapy causes high rates of mortality among several cancer patients." The authors should clarify this term by noting a specific percentage.
3. Lines 148-149: the authors should indicate the source confirming this fact, since, obviously, this is not the authors' own conclusion. In general, I would like to recommend that the authors supplement the last paragraph of the third chapter with links to relevant sources of information. In particular, I could invite them to quote the following works in the first sentence:
- EGCG-induced selective death of cancer cells through autophagy-dependent regulation of the p62-mediated antioxidant survival pathway. H.W. Lee, J.H. Choi, D. Seo et al. Biochimica et biophysica acta. Molecular cell research. 2024. 1871, 3, 119659. https://doi.org/10.1016/j.bbamcr.2024.119659
- New Conjugates of Daunorubicin with Sesquiterpene Lactones and Their Biological Activity. S.A. Pukhov, A.V. Semakov, A.A. Globa et al. ChemistrySelect. 2021. 6, 32. 8446-8451. https://doi.org/10.1002/slct.202102244
- Curcumin Nicotinate Selectively Induces Cancer Cell Apoptosis and Cycle Arrest through a P53-Mediated Mechanism. Y.C. He, L. He, R. Khoshaba et al. Molecules. 2019. 24, 22. 4179. https://doi.org/10.3390/molecules24224179
- Conjugates of 3,5-Bis(arylidene)-4-piperidone and Sesquiterpene Lactones Have an Antitumor Effect via Resetting the Metabolic Phenotype of Cancer Cells. M.E. Neganova, Yu.R. Aleksandrova, E.V. Sharova et al. Molecules. 2024. 29, 12. 2765. https://doi.org/10.3390/molecules29122765
4. Lines 195, 204, 208, 216, 226, 244, 313, 352, 413, 492, 493, 514, 545, 580, 586, 591, 706: " Artemisia annua, Momordica charantia L., Rhizoma curcumae, Berberis vulgaris, Leguminosae, Catharantus roseus, Curcuma longa, Magnolia officinalis". Latin names of species/families/genera of plants are italicized, the authors should check this flaw in the entire text of the manuscript.
5. Similar to the remark above: in vitro, in vivo, in situ are italicized.
6. Figure 2 looks cropped. Should it be like that? If not, then the authors should correct this flaw.
7. I would also like to recommend that authors standardize the design of references to literary sources in a single style. Currently, the design of references to literary sources differs from each other.
Author Response
Reviewer #2
We thank the reviewer for taking the time to review our manuscript and for the valuable comments. We present herewith our point-to-point rebuttal in response to the comments:
Reviewer comments:
- Lines 123-124: "problems of the mouth"; in my opinion, an incorrect phrase, perhaps it is better to use "problems with the oral cavity" or "problems with the oral mucosa".
Author reply: We thank the reviewer for the comment. The said correction has been made.
- Lines 144-146: "The delay in cancer diagnosis combined with non-responsive therapy causes high rates of mortality among several cancer patients." The authors should clarify this term by noting a specific percentage.
Author reply: The said correction has been made and the specific percentage has been provided.
- Lines 148-149: the authors should indicate the source confirming this fact, since, obviously, this is not the authors' own conclusion. In general, I would like to recommend that the authors supplement the last paragraph of the third chapter with links to relevant sources of information. In particular, I could invite them to quote the following works in the first sentence:
- EGCG-induced selective death of cancer cells through autophagy-dependent regulation of the p62-mediated antioxidant survival pathway. H.W. Lee, J.H. Choi, D. Seo et al. Biochimica et biophysica acta. Molecular cell research. 2024. 1871, 3, 119659. https://doi.org/10.1016/j.bbamcr.2024.119659
- New Conjugates of Daunorubicin with Sesquiterpene Lactones and Their Biological Activity. S.A. Pukhov, A.V. Semakov, A.A. Globa et al. ChemistrySelect. 2021. 6, 32. 8446-8451. https://doi.org/10.1002/slct.202102244
- Curcumin Nicotinate Selectively Induces Cancer Cell Apoptosis and Cycle Arrest through a P53-Mediated Mechanism. Y.C. He, L. He, R. Khoshaba et al. Molecules. 2019. 24, 22. 4179. https://doi.org/10.3390/molecules24224179
- Conjugates of 3,5-Bis(arylidene)-4-piperidone and Sesquiterpene Lactones Have an Antitumor Effect via Resetting the Metabolic Phenotype of Cancer Cells. M.E. Neganova, Yu.R. Aleksandrova, E.V. Sharova et al. Molecules. 2024. 29, 12. 2765. https://doi.org/10.3390/molecules29122765
Author reply: All the above references have been cited in Section 3 last paragraph as kindly specified by the reviewer.
- Lines 195, 204, 208, 216, 226, 244, 313, 352, 413, 492, 493, 514, 545, 580, 586, 591, 706: " Artemisia annua,Momordica charantia L., Rhizoma curcumae, Berberis vulgaris, Leguminosae, Catharantus roseus, Curcuma longa, Magnolia officinalis". Latin names of species/families/genera of plants are italicized, the authors should check this flaw in the entire text of the manuscript.
Author reply: We thank the reviewer for this comment. All the above corrections have been made at the appropriate places throughout the manuscript.
- Similar to the remark above: in vitro, in vivo, in situare italicized.
Author reply: the words in vitro, in vivo, in situ are italicized as per the reviewer’s valuable comment.
- Figure 2 looks cropped. Should it be like that? If not, then the authors should correct this flaw.
Author reply: The complete figure has been provided.
- I would also like to recommend that authors standardize the design of references to literary sources in a single style. Currently, the design of references to literary sources differs from each other.
Author reply: Thanks for your valuable comment. All the references have been cited using EndNote software.

Reviewer 3 Report
Comments and Suggestions for Authors
This article reviews the anticancer activities compounds from plant phytochemicals, that focus on targeted cancer prophylaxis and therapy. Also addressed are plant-derived anti-cancer vaccines, nanoparticles, monoclonal antibodies and immunotherapies. However, the content of the article is too lengthy and loses the original focus. Please correct it according to the following comments.
Major comments
1. 1. The article is too long. It should be shortened, and can be condensed into several key narratives.
2. Please provide a graphical abstract or a schematic diagram for quick understanding of the manuscript.
3. The authors can provide more constructive or practical reference methods for cancer therapy based on the characteristics of plants.
4. As a review paper, various PVNPs therapeutic mechanisms should be compared in detail.
5. The preparation or extraction methods of various plant anti-cancer molecules can be integrated into diagrams or tables, and whether further purification is needed can be discussed.
Author Response
- The article is too long. It should be shortened and can be condensed into several key narratives.
Author reply: Sections have been removed and key points have been provided (please check the version with “Track changes” enabled).
- Please provide a graphical abstract or a schematic diagram for quick understanding of the manuscript.
Author reply: We thank the reviewer for the valuable comment. A graphical abstract has been provided reflecting the theme of the manuscript.
- The authors can provide more constructive or practical reference methods for cancer therapy based on the characteristics of plants.
Author reply: We thank the reviewer for this comment. A section has been included addressing cancer therapy based on plant characteristics.
- As a review paper, various PVNPs therapeutic mechanisms should be compared in detail. Please provide a table comparing these mechanisms.
Author reply: We added a table comparing the various PVNPs therapeutic mechanisms.
- The preparation or extraction methods of various plant anti-cancer molecules can be integrated into diagrams or tables, and whether further purification is needed can be discussed.
Author reply: A table has been included to address the extraction methods of phytochemicals in accordance with the reviewer’s valuable comments.

Round 2
Reviewer 1 Report
Comments and Suggestions for Authors
Figures 3 and 4: figure captions need to be checked.